# STYLESTREAM:
# REAL-TIME ZERO-SHOT VOICE STYLE CONVERSION

## ABSTRACT

Voice style conversion aims to transform an input utterance to match a target speaker's timbre, accent, and emotion. A central challenge is disentangling linguistic content from style attributes. While prior work has investigated this disentanglement, the conversion quality remains suboptimal. Moreover, no existing work addresses real-time voice style conversion. To address these limitations, we propose StyleStream, the first streamable zero-shot voice style conversion system that achieves state-of-the-art conversion performance. StyleStream mainly consists of two components: a destylizer, which removes style attributes (timbre, accent and emotion) while retaining linguistic content, and a stylizer, which is a diffusion transformer (DiT) that reintroduces style conditioned on the target speech. Content–style disentanglement is enforced in the destylizer through two mechanisms: (i) automatic speech recognition (ASR) loss that provides text-level supervision, and (ii) a finite scalar quantization (FSQ) module with a compact codebook of size 45, which serves as a strong information bottleneck. The continuous representations preceding the FSQ layer are treated as the content features. By combining chunked-causal attention masking with a non-autoregressive architecture, StyleStream enables real-time voice style conversion with an end-to-end latency of 1 second. Samples can be found here.

## 1 INTRODUCTION

Zero-shot voice style conversion seeks to modify an input utterance so that it reflects the timbre, accent, and emotion of an unseen target speaker (collectively defined in this work as *voice style*) while preserving the original linguistic content, using only a few seconds of reference speech. Although zero-shot voice style cloning has been extensively studied in the text-to-speech (TTS) domain (Le et al., 2023; Eskimez et al., 2024; Chen et al., 2024; Wang et al., 2025c; Anastassiou et al., 2024; Du et al., 2024a;b), progress on zero-shot voice style conversion remains limited. In TTS, the linguistic content is taken directly from text, which is fully disentangled from the target speech that provides voice style. In contrast, zero-shot voice style conversion requires the model to first extract the content features from the source utterance, and then generate speech in the target style. This content extraction relies on clean content-style disentanglement, which is the core challenge in speech-to-speech conversion.

Existing strategies for disentanglement include information bottleneck (Qian et al., 2019; 2020; Ju et al., 2024; Du et al., 2024a;b; Zhang et al., 2025b), signal perturbation (Choi et al., 2021; 2022; Qian et al., 2022; Chan et al., 2022), mutual information minimization (Wang et al., 2021; Zhu et al., 2023; Tjandra et al., 2021; Du et al., 2022), etc. While these approaches achieve reasonable content–timbre disentanglement, the extracted features often remain entangled with accent and emotion. For example, CosyVoice 2 (Du et al., 2024b) combines information bottlenecks with ASR for supervised disentanglement, yet its "semantic tokens" still encode considerable accent and emotion information due to the large codebook size (6561), as demonstrated in Section 5.3. The current state-of-the-art framework, Vevo (Zhang et al., 2025b), employs VQ-VAE (Van Den Oord et al., 2017) to quantize HuBERT (Hsu et al., 2021) features with a compact codebook of size 32, producing discrete content tokens. Although Vevo demonstrates good conversion performance, its purely self-supervised training provides no guarantee on what information is discarded by the bottleneck, and linguistic content is prone to degradation during quantization, leading to suboptimal intelligibility and conversion quality (see Section 5). Beyond disentanglement, another open challenge is real-

Figure 1: System overview of StyleStream. The destylizer extracts content features disentangled from style, and the stylizer generates speech that preserves the source linguistic content while adopting the target timbre, accent, and emotion.

time application: while real-time voice conversion (timbre only) has been widely studied (Liu et al., 2025; Yang et al., 2024; Wang et al., 2024b;c), there is no existing work that addresses real-time voice style conversion (timbre, accent and emotion), leaving an important gap.

Motivated by these challenges, we present StyleStream, a novel framework that, for the first time, supports zero-shot real-time voice style conversion across timbre, accent, and emotion, achieving state-of-the-art performance. As shown in Figure 1, the destylizer first extracts content features that are disentangled from style information. The stylizer then takes these content features and, conditioned on the target speech, generates the stylized mel-spectrogram, which is subsequently converted into waveform by the vocoder.

To achieve better content–style disentanglement, inspired by CosyVoice 2 (Du et al., 2024b), we train the destylizer with an ASR loss and apply finite scalar quantization (FSQ) (Mentzer et al., 2023) as an information bottleneck, but unlike CosyVoice 2, where a large codebook size (6561) is used, we constrain the codebook size to 45. This combination of text supervision and a compact codebook enables cleaner disentanglement. The choice of content features is also crucial: rather than using the discrete codes directly, we adopt the continuous representations immediately preceding the FSQ layer as content features, following the intuition of SoftVC (Van Niekerk et al., 2022). As shown in Section 5.4, this design is another critical factor for effective disentanglement.

For the stylizer, we train a diffusion transformer (DiT) (Peebles & Xie, 2022) with a spectrogram inpainting objective, similar to (Le et al., 2023; Eskimez et al., 2024; Chen et al., 2024). Since no autoregressive modules are involved, the input and output lengths remain identical, which makes streaming straightforward and avoids lag or overlap caused by input-output length mismatches.

In summary, our contributions include:

- We introduce StyleStream, the first real-time voice style conversion system with an end-to-end latency of 1 second.
- By combining ASR loss and a compact quantization codebook, the destylizer achieves cleaner content-style disentanglement compared to existing methods.
- Trained on 50k hours of English data, StyleStream delivers state-of-the-art conversion quality, substantially improving accent and emotion similarity over prior work.

## 2 RELATED WORK

### 2.1 VOICE STYLE CLONING

A large body of work in TTS focuses on voice style cloning, where models aim to reproduce a target speaker's timbre, accent, and emotion from limited reference speech. Modern TTS approaches to voice style cloning can be broadly categorized into two classes based on their architecture: (1) Non-autoregressive: approaches that mainly utilize diffusion transformer models, trained with a feature inpainting objective (Le et al., 2023; Vyas et al., 2023; Eskimez et al., 2024; Chen et al., 2024; Lee et al., 2024; Zhu et al., 2025; Wang et al., 2025c); (2) Autoregressive: approaches that use text as a sequence prefix and train a decoder-only transformer with either a next-speech-token

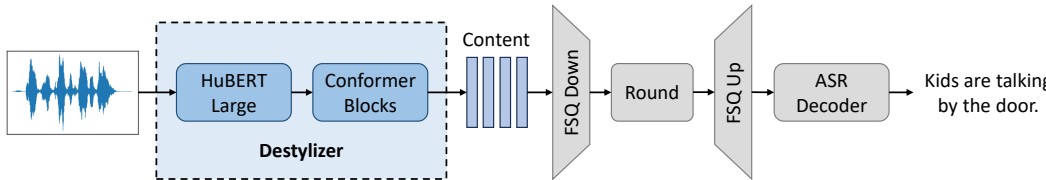

Figure 2: Destylizer architecture. The destylizer, as part of the ASR encoder, is trained with a sequence-to-sequence ASR loss. The continuous representations immediately before the FSQ module are taken as content features.

prediction objective (Wang et al., 2023; Anastassiou et al., 2024; Du et al., 2024a;b; Guo et al., 2024; Deng et al., 2025; Zhou et al., 2025; Zhang et al., 2025a; Wang et al., 2025b) or a next-continuous-feature prediction objective (Meng et al., 2024; Jia et al., 2025; Peng et al., 2025; Wang et al., 2025a). Although high-fidelity voice style cloning has been achieved in the TTS domain, a considerable gap remains in speech-to-speech conversion, since TTS systems directly use text as content features, which are naturally disentangled from style, whereas speech-to-speech conversion requires extracting clean content features through content-style disentanglement.

## 2.2 SPEECH CONTENT DISENTANGLEMENT

A key difficulty in speech-to-speech conversion lies in isolating linguistic content from the input speech. Research on speech content disentanglement generally falls into a few main categories: (1) Information bottleneck: one line of work introduces a quantization layer (Polyak et al., 2021; Van Niekerk et al., 2022; Huang et al., 2021; Wu & Lee, 2020; Xue et al., 2024) or a low-dimensional hidden representation (Tang et al., 2023; Cho et al., 2024; Liu et al., 2024), often combined with proxy tasks such as ASR (Zhou et al., 2022; Du et al., 2024a;b; 2025) or autoencoding (Qian et al., 2019; 2020; Zhang et al., 2025b), to encourage the separation of content from style; (2) Signal perturbation: another family of methods (Choi et al., 2021; 2022; Qian et al., 2022; Liu, 2024) modifies style-related cues in the signal (e.g., pitch randomization, formant shifting) while preserving linguistic content, and trains the model to produce similar representations for the original and perturbed inputs; (3) Training loss design: alternative approaches incorporate objectives such as GAN loss (Kameoka et al., 2018; Kaneko et al., 2019; Zhou et al., 2021a), mutual information loss (Wang et al., 2021; Zhu et al., 2023; Tjandra et al., 2021; Du et al., 2022; Lian et al., 2022), or gradient reversal loss (Ju et al., 2024; Łajszczak et al., 2024) to explicitly promote disentanglement. However, content features extracted by these methods often still leak accent and emotion information, which degrades the quality of voice style conversion.

## 2.3 REAL-TIME VOICE CONVERSION

The field of real-time voice conversion has seen rapid progress in both quality and latency (Wang et al., 2024b;c; Yang et al., 2024; Liu et al., 2025; Ning et al., 2023; 2024a;b). For example, RT-VC (Liu et al., 2025) achieves state-of-the-art real-time zero-shot voice conversion quality with a CPU latency of only 61.4ms. However, no existing work has addressed the problem of real-time zero-shot voice style conversion, where the goal is to modify not only timbre but also accent and emotion. This gap motivates our work, where we present StyleStream, the first system for real-time zero-shot voice style conversion.

## 3 METHOD

### 3.1 DESTYLIZER: CONTENT-STYLE DISENTANGLEMENT

To achieve clean content-style disentanglement, we introduce the destylizer (Figure 2). It consists of a frozen HuBERT-Large encoder (Hsu et al., 2021) followed by several conformer blocks (Gulati et al., 2020). Similar to CosyVoice 2 (Du et al., 2024b), the destylizer and the FSQ module together form the ASR encoder. The continuous output features of the destylizer, denoted as content features $f_c$, are first projected down to a $D$-dimensional space, and each dimension is quantized to the nearest

integer in the range $[-K_i, K_i]$, where

$$V_i = 2K_i + 1, \quad K_i \in \mathbb{Z}_{>0} \tag{1}$$

denotes the number of codes for the $i$-th dimension. The quantized low-dimensional features are then projected back to the original dimension and passed to the ASR decoder to predict text character tokens. The whole pipeline is trained end-to-end with a sequence-to-sequence ASR loss. At inference time, we use only the destylizer to extract disentangled content features, operating at a sampling rate of 50 Hz.

There are three keys to successful content-style disentanglement:

**Text Supervision** Unlike Vevo (Zhang et al., 2025b), whose content tokenizer is trained in a purely self-supervised manner, we follow CosyVoice 2 and impose text supervision for training the destylizer. This directs linguistic content through the FSQ bottleneck, while style is suppressed since it does not aid ASR prediction and the bottleneck provides limited capacity. In contrast, Vevo relies on a reconstruction objective with no explicit cue for separating content from style, causing style leakage and partial loss of linguistic content, which degrades disentanglement quality, as shown in Section 5.

**Compact Codebook** Unlike CosyVoice 2, where a large codebook size (6561) is used, we use a much smaller codebook to enforce a narrower information bottleneck, following the intuition of AUTOVC (Qian et al., 2019). Specifically, the vocabulary size $V$ for FSQ levels $[V_1, V_2, ..., V_D]$ is given by:

$$V = \prod_{i=1}^{D} V_i \tag{2}$$

Adjusting $D$ and $V_i$ controls the bottleneck width. As demonstrated in Section 5.3, CosyVoice 2's "semantic tokens" remain largely entangled with style information, whereas our destylizer provides much cleaner content features due to a compact codebook.

**Continuous Pre-quantization Features** Unlike Vevo (Zhang et al., 2025b) or CosyVoice 2 (Du et al., 2024b), which use discrete tokens as content, we instead adopt the continuous representations immediately before FSQ, inspired by SoftVC (Van Niekerk et al., 2022). SoftVC demonstrated that such "soft units" strike an effective balance between disentanglement and content preservation. Moreover, as shown in Section 5.3, our continuous features contain even less style information than the discrete tokens of Vevo and CosyVoice 2, while using FSQ indices directly as content leads to unintelligible speech.

## 3.2 STYLIZER: STYLIZED ACOUSTIC MODELING

After obtaining disentangled content features, the stylizer generates mel-spectrograms conditioned on the target style. As shown in Figure 3, it consists of two components:

**Diffusion Transformer** We use a diffusion transformer (DiT) (Peebles & Xie, 2022) as the backbone of the stylizer, as it has demonstrated strong performance in in-context voice style cloning (Le et al., 2023; Eskimez et al., 2024; Chen et al., 2024).The stylizer is trained with a spectrogram in-painting objective: given a temporal binary mask $m$, the model reconstructs the masked segment $m \odot x_1$ of a mel-spectrogram $x_1 \in \mathbb{R}^{F \times T}$, conditioned on the unmasked context $(1 - m) \odot x_1$, the content features $f_c \in \mathbb{R}^{D_c \times T}$, and a style embedding $e$ (described below), where $\odot$ denotes elementwise multiplication.

Concretely, the noisy mel-spectrogram $x_t$, context $(1 - m) \odot x_1$ and content features $f_c$ are concatenated along the channel dimension to form the DiT input. To ensure compatibility, the mel-spectrogram is calculated at the same rate as $f_c$ (50 Hz). The flow time step $t \sim \mathcal{U}[0, 1]$ is embedded with a sinusoidal positional encoding, and added to the style embedding, which is then integrated into DiT via adaLN-zero (Peebles & Xie, 2022).

During training, we adopt the conditional flow matching (CFM) loss with an Optimal Transport (OT) path formulation. Let $x_1 \sim q(x)$ denote the ground-truth mel-spectrogram drawn from the data distribution, and $x_0 \sim p(x) = \mathcal{N}(0, I)$ be the standard normal prior. We sample a time step $t \sim \mathcal{U}[0, 1]$ and define the OT flow path as $\psi_t(x) = (1 - t)x + tx_1$. Consequently, the noisy mel-spectrogram state at time $t$ is generated as $x_t = \psi_t(x_0)$. The training objective is to minimize

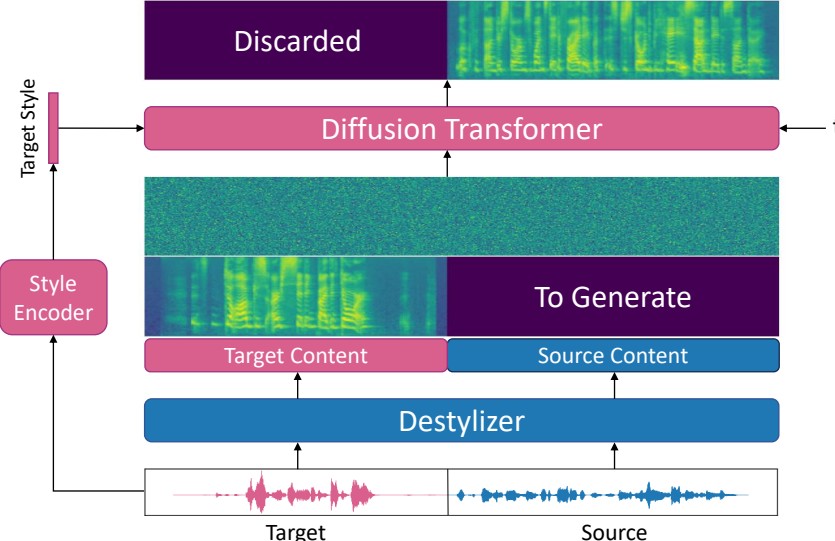

Figure 3: Stylizer architecture. The Stylizer contains a style encoder and a diffusion transformer.

the difference between the predicted vector field $v_\theta$ and the target velocity, computed strictly on the masked regions:

$$\mathcal{L}_{\mathbf{CFM}}(\theta) = \mathbb{E}_{t,x_1,x_0,m} \left\| m \odot \left( v_\theta(\psi_t(x_0), t; f_c, (1-m) \odot x_1, e) - \frac{d}{dt}\psi_t(x_0) \right) \right\|^2 \quad (3)$$

where $m$ is the binary mask indicating the generation target, $f_c$ represents the content features, and $e$ denotes the style embedding.

During inference, we concatenate the content features of the target and source utterances along the time axis. The target utterance provides the mel-spectrogram context and the style embedding, while the source region is masked and generated by the stylizer. To balance diversity with fidelity, we employ Classifier-Free Guidance (CFG):

$$v_{\theta,\mathbf{CFG}} = v_\theta(\psi_t(x_0), t; c) + \alpha(v_\theta(\psi_t(x_0), t; c) - v_\theta(\psi_t(x_0), t; \phi)) \quad (4)$$

where $\phi$ denotes null condition, $c$ denotes all conditions, including $f_c, (1-m) \odot x_1, e$, and $\alpha$ is the CFG strength.

**Style Encoder** For style conditioning, we introduce a style encoder to capture global style attributes. Its architecture follows WavLM-TDNN[1]: representations from WavLM layers (Chen et al., 2022) are aggregated using learnable coefficients, and the aggregated features are then passed to a Time Delay Neural Network (TDNN) (Desplanques et al., 2020). Attentive statistics pooling (Okabe et al., 2018) is applied to obtain the final style embedding, which is fused into DiT through adaLN-zero. The style encoder is trained jointly with the DiT in an end-to-end manner.

### 3.3 VOCODER

We trained a causal vocoder to synthesize 16 kHz speech from 50 Hz mel-spectrograms for real-time inference. The design follows Vocos (Siuzdak, 2023), with all convolution layers replaced by causal convolution. The training objective combines GAN loss, reconstruction loss, and feature matching loss, using the same configurations as Vocos. The original Vocos checkpoint[2] is used as a warm start.

---

[1]https://huggingface.co/microsoft/wavlm-base-plus-sv
[2]https://github.com/gemelo-ai/vocos

### 3.4 REAL-TIME DESIGN

To enable streaming, we apply a chunked-causal attention mask to both the destylizer and stylizer, where each chunk attends to its own features and all preceding chunks, but not to future chunks. In the destylizer, the HuBERT layers are unfrozen and made chunked-causal, and all convolution layers are converted to causal. The streaming destylizer is initialized from its non-streaming checkpoint and trained with a mean squared error (MSE) distillation loss against the non-streaming teacher's content features. Once trained, the streaming stylizer—warm-started from its non-streaming counterpart—is trained on top of the streaming destylizer's outputs.

For real-time inference, we maintain a fixed-length target utterance and a ring buffer of input chunks of content features. This provides sufficient past context for the stylizer while keeping latency manageable. The end-to-end latency $L$ is calculated as:

$$L = t_{\text{chunksize}} + t_{\text{proc}} \tag{5}$$

where $t_{\text{chunksize}}$ is the input speech chunk size and $t_{\text{proc}}$ is the processing time per chunk. As long as $t_{\text{proc}} < t_{\text{chunksize}}$, the model is streamable.

## 4 EXPERIMENTAL SETUP

### 4.1 DATASET

For the training data, the stylizer is trained using the English portion of Emilia (He et al., 2024), which contains 50k hours of diverse in-the-wild English speech. The destylizer is trained on a combined training set from LibriTTS (Zen et al., 2019), MSP-Podcast (Lotfian & Busso, 2017), and GLOBE (Wang et al., 2024a), totaling approximately 1300 hours of speech that covers a wide range of speakers, emotions and accents. We refer to this combined dataset as LMG. LMG is also used for training in the ablation studies (Sections 5.3, 5.4).

For evaluation, we randomly sample 300 source utterances from a combination of the Emotion Speech Dataset (ESD) (Zhou et al., 2021b), GLOBE-test, and LibriTTS-test-clean. We also select 10 target utterances from ESD, RAVDESS (Livingstone & Russo, 2018), GLOBE-test, and L2-ARCTIC (Zhao et al., 2018), covering 5 emotions (happy, angry, sad, fearful, calm) and 5 accents (British, American, Indian, Arabic, Mandarin). This results in $300 \times 10 = 3000$ source-target pairs, which we denote as StyleStream-Test. All the objective and subjective evaluations in Section 5.1 are conducted on StyleStream-Test. Moreover, to evaluate content-style disentanglement (Section 5.3), we train style classifiers on top of the content features extracted from different models. The speaker classifier is trained on the VoxCeleb training set (Nagrani et al., 2020) and evaluated on its test set, which contains 1,251 speakers. The accent classifier is trained on L2-ARCTIC, which includes 6 accents; 6 speakers (one per accent) are held out for testing, and the remaining 18 speakers are used for training. The emotion classifier is trained on EmoV-DB (Adigwe et al., 2018), which covers 5 emotions; 2 speakers (each with all 5 emotions) are held out for testing, and the remaining 18 speakers are used for training.

### 4.2 TRAINING

**Destylizer** We take the 18th layer of HuBERT-Large-ASR[3] as input to six conformer blocks, followed by an FSQ module and four transformer decoder layers (hidden size 768, FFN size 3072). ALiBi positional encoding (Press et al., 2021) is adopted to reduce long-context reliance, making the model suitable for length extrapolation and real-time inference. FSQ levels are set to [5,3,3], yielding a compact codebook of 45 codes. The model is trained for 100k steps on 8 NVIDIA RTX A6000 GPUs (batch size 32) using AdamW (Loshchilov & Hutter, 2017) with a peak learning rate of 1e-4, 4k warm-up steps, and cosine annealing. The streaming variant uses a chunk size of 600 ms but otherwise follows the same training configuration.

**Stylizer** The stylizer consists of 16 transformer layers (hidden size 768, FFN size 3072). Speech is sampled at 16 kHz and converted to 100-bin mel-spectrograms with a hop size of 320, producing a 50 Hz frame rate. During training, $70 - 100\%$ of mel-spectrogram frames are randomly masked

---

[3]https://huggingface.co/facebook/hubert-large-ls960-ft

Table 1: Results for zero-shot voice style conversion. The best results are shown in **bold**, and the second best results are underlined.

| Model | WER (%) ↓ | S-SIM ↑ | A-SIM ↑ | E-SIM ↑ | NMOS ↑ | A-SMOS ↑ | E-SMOS ↑ | S-SMOS ↑ |
|---|---|---|---|---|---|---|---|---|
| Ground Truth | 3.8 | — | — | — | 3.71±.11 | — | — | — |
| FACodec (Ju et al., 2024) | 15.5 | 0.763 | 0.408 | 0.668 | 2.55±.13 | 2.02±.13 | 2.76±.14 | 2.98±.14 |
| CosyVoice 2.0 (Du et al., 2024b) | 9.5 | 0.794 | 0.450 | 0.655 | **3.47**±.11 | 2.26±.13 | 2.58±.15 | 3.25±.14 |
| SeedVC v2[4] | 21.7 | 0.766 | 0.549 | 0.688 | 2.65±.12 | 3.34±.12 | 3.21±.13 | 3.11±.13 |
| Vevo (Zhang et al., 2025b) | 17.5 | 0.818 | 0.596 | 0.712 | 3.38±.12 | 3.93±.11 | 3.49±.13 | 3.76±.13 |
| Vevo 1.5 (Li et al., 2025) | 20.5 | 0.798 | 0.554 | 0.683 | 3.10±.12 | 3.15±.14 | 3.15±.14 | 3.34±.14 |
| StyleStream (streaming) | 15.3 | **0.855** | 0.635 | 0.803 | 3.34±.13 | 4.28±.09 | 4.37±.09 | 4.29±.10 |
| StyleStream (offline) | **9.2** | 0.852 | **0.640** | **0.827** | 3.42±.12 | **4.32**±.09 | **4.42**±.09 | **4.36**±.10 |

for inpainting. To support CFG inference, content features are dropped with probability 0.2, while context spectrograms and style embeddings are dropped with probability 0.3. Training uses 6s segments for 400k steps on 8 A6000 GPUs (batch size 64), with AdamW at a peak learning rate of 1e-4, 2k warm-up steps, and cosine annealing. The streaming stylizer also uses a 600 ms chunk size under the same configuration. Throughout the experiments, the CFG strength is fixed at 2 and the Number of Function Evaluations (NFE) is set to 16, utilizing the standard Euler sampling method.

**Vocoder** We follow the design choice of Vocos (Siuzdak, 2023), but modify the convolution layers in ConvNext (Liu et al., 2022) blocks into causal convolutions. Training is initialized from the official checkpoint and performed on the LibriTTS training set for 100k steps, using 2 A6000 GPUs with a batch size of 64 and 2s input segments. The mel-spectrogram setup matches that of the stylizer, and all remaining training configurations follow Vocos.

## 4.3 BASELINES

We compare StyleStream against other voice conversion systems, including CosyVoice 2.0 (Du et al., 2024b) and FACodec (Ju et al., 2024), as well as state-of-the-art voice style conversion models Vevo (Zhang et al., 2025b) and Vevo 1.5 (Li et al., 2025). We also include SeedVC v2[4], an upgraded version of SeedVC (Liu, 2024) that supports voice style conversion. More details about the baselines can be found in Appendix A.

## 4.4 METRICS

For objective evaluation, we report word error rate (WER) as a measure of intelligibility, computed with Whisper-large-v3 (Radford et al., 2023). To evaluate style similarity, we extract embeddings for speaker[5], accent[6] (Zuluaga-Gomez et al., 2023), and emotion[7] (Ma et al., 2023), and compute cosine similarity between generated and target speech, yielding speaker similarity (S-SIM), accent similarity (A-SIM), and emotion similarity (E-SIM). For subjective evaluation, we conduct Mean Opinion Score (MOS) tests on a 1–5 scale, including naturalness (N-MOS) of the converted speech and similarity MOS (SMOS) for speaker (S-SMOS), accent (A-SMOS), and emotion (E-SMOS) relative to the target. See Appendix B for additional details on MOS and SMOS.

## 5 RESULTS

### 5.1 ZERO-SHOT VOICE STYLE CONVERSION

The main results are shown in Table 1. For intelligibility, the offline StyleStream achieves the lowest WER (9.2%), outperforming CosyVoice 2.0 and Vevo, while the streaming variant remains comparable to Vevo despite being chunked-causal. This demonstrates that the destylizer effectively preserves linguistic content during disentanglement. For speaker similarity, our models achieve the highest S-SIM and S-SMOS, highlighting stronger perceptual timbre similarity. For accent transfer,

---

[4] https://github.com/Plachtaa/seed-vc

[5] https://github.com/resemble-ai/Resemblyzer

[6] https://huggingface.co/Jzuluaga/accent-id-commonaccentecapa

[7] https://github.com/ddlBoJack/emotion2vec

Table 2: Chunk size vs. processing time on RTX 4060 and RTX A6000. Entries show mean $\pm$ standard deviation over runs.

| Chunk size (ms) | RTX 4060 (s) | RTX A6000 (s) |
|---|---|---|
| 100 | $0.537\pm.053$ | $0.427\pm.013$ |
| 200 | $0.574\pm.059$ | $0.429\pm.009$ |
| 300 | $0.615\pm.057$ | $0.432\pm.010$ |
| 400 | $0.647\pm.090$ | $0.441\pm.008$ |
| 500 | $0.653\pm.051$ | $0.431\pm.009$ |
| 600 | $0.668\pm.048$ | $0.429\pm.006$ |
| 700 | $0.673\pm.041$ | $0.429\pm.002$ |
| 800 | $0.673\pm.040$ | $0.431\pm.004$ |

Table 3: Chunk size vs. quality tradeoff

| Chunk size (ms) | WER (%) $\downarrow$ | S-SIM $\uparrow$ | A-SIM $\uparrow$ | E-SIM $\uparrow$ | UTMOS $\uparrow$ |
|---|---|---|---|---|---|
| 200 | 19.8 | 0.797 | 0.586 | 0.704 | $2.33\pm.02$ |
| 400 | 15.0 | 0.806 | 0.597 | 0.720 | $2.69\pm.03$ |
| 600 | 13.6 | 0.811 | 0.602 | 0.734 | $2.85\pm.03$ |
| 800 | 12.7 | 0.816 | 0.607 | 0.745 | $2.94\pm.03$ |
| 1000 | 12.6 | 0.820 | 0.613 | 0.751 | $3.00\pm.03$ |

the offline variant obtains the best A-SIM, and both variants reach significantly higher A-SMOS, confirming more accurate accent conversion. For emotion, StyleStream achieves the highest similarity to the target emotion: the offline model leads in E-SIM, and both variants substantially surpass baselines in E-SMOS.

## 5.2 STREAMING ANALYSIS

**Latency**    We measure the end-to-end streaming latency on a single NVIDIA RTX A6000 GPU. Using NFE=16, a chunk size of $t_{\mathrm{chunksize}} = 600$ms, a 5s target segment, and a 5s content ring buffer, with the target style embedding pre-extracted for inference, the average processing time per chunk is $t_{\mathrm{proc}} = 412.7$ms. Since $t_{\mathrm{proc}} < t_{\mathrm{chunksize}}$, streaming is feasible. By Equation 5, the resulting end-to-end latency is $L = 600 + 412.7 = 1012.7$ms.

**Chunksize-Latency Analysis**    We benchmark the processing time ($t_{\mathrm{proc}}$) across varying chunk sizes on both a server-grade NVIDIA A6000 and a consumer-grade RTX 4060 Laptop GPU (NFE fixed at 16). As shown in Table 2, $t_{\mathrm{proc}}$ on the A6000 remains effectively constant ($\approx 0.43$s) when varying the chunk size from 100ms to 800ms. This indicates that the DiT inference is dominated by fixed kernel launch overheads rather than computational saturation on high-end hardware. In contrast, the RTX 4060 exhibits a linear increase in processing time with chunk size, reflecting a compute-bound regime characteristic of consumer-grade deployment.

**Chunksize-Quality Trade-off**    To decouple the effects of context length from training-inference mismatch, we analyze the chunksize-quality trade-off by simulating streaming inference using the offline checkpoint across chunk sizes ranging from 200ms to 1000ms (Table 3). We observe a monotonic improvement in both intelligibility (WER) and style similarity as chunk size increases. This confirms that extended temporal context is essential for capturing accent and emotion and minimizing discontinuities at chunk boundaries.

## 5.3 ABLATIONS ON CONTENT FEATURES

In this section, we run several ablations on the different content features in order to measure how the destylizer contributes to the downstream performance. To this end, we train the stylizer from scratch on top of the destylizer and other existing speech content features and measure S-SIM, A-SIM, E-SIM, and UTMOS Saeki et al. (2022), a machine-evaluated MOS. Specifically, we compare against the continuous pre-quantization features and discrete representations from the current SOTA in voice style conversion, Vevo, as well as the raw 18th-layer features of HuBERT-Large-ASR, which are incorporated as the first part of the destylizer. All the stylizers are trained on the LMG

Table 4: Ablation on the stylizer performance trained on different content features.

| Content Features | WER (%) ↓ | S-SIM ↑ | A-SIM ↑ | E-SIM ↑ | UTMOS ↑ |
|---|---|---|---|---|---|
| HuBERT-Large-ASR 18th | 12.5 | 0.731 | 0.389 | 0.627 | 3.28±.020 |
| Vevo Continuous | 12.6 | 0.767 | 0.420 | 0.660 | 2.92±.029 |
| Vevo Indices | 25.6 | 0.757 | 0.564 | 0.657 | 2.65±.036 |
| Destylizer (offline) | **10.7** | **0.837** | **0.626** | **0.733** | 3.22±.029 |

Table 5: Style classification accuracy across different content features. "Acc." stands for accuracy.

| Content Features | Accent Acc. ↓ | Emotion Acc. ↓ | Speaker Acc. ↓ |
|---|---|---|---|
| HuBERT-Large-ASR 18th | 78.00% | 85.60% | 86.60% |
| Vevo Continuous | 68.93% | 78.73% | 64.76% |
| Vevo Indices | 55.60% | 53.40% | 23.00% |
| CosyVoice 2.0 Indices | 50.60% | 56.20% | 25.00% |
| Destylizer (offline) | 43.50% | 47.60% | 3.50% |
| Destylizer (streaming) | **33.80%** | **37.90%** | **3.01%** |

dataset detailed in Section 4.1, with the same training configurations in Section 4.2, except that training is limited to 100k steps.

The results are reported in Table 4. Stylizers trained on destylizer features achieve the best performance in WER, S-SIM, A-SIM, and E-SIM, while remaining competitive on UTMOS. In contrast, the HuBERT-based stylizer achieves the lowest A-SIM and E-SIM, indicating poor style conversion performance. For Vevo, using discrete indices yields much higher A-SIM than its continuous features, but this comes at a substantial cost to content preservation, as reflected by the highest WER.

We hypothesize that the differences in downstream stylizer performance are closely linked to the degree of content–style disentanglement. When disentanglement is poor, style information leaked into the content features may be exploited by the stylizer during training, causing it to reconstruct speech with source-style attributes. As a result, inference fails to faithfully reflect the target style, leading to degraded style fidelity. To validate this, we conduct a series of style classification experiments to assess how much accent, emotion, and speaker information remain in the extracted content features.

To this end, we train an ECAPA-TDNN (Desplanques et al., 2020) classifier on top of each set of content features to predict speaker, accent, and emotion (see Appendix C for details). The goal is to approach near-random classification accuracy, since these style attributes are precisely what the destylizer is designed to remove. Results are shown in Table 5. Both variants of the destylizer consistently achieve the strongest disentanglement across all three tasks. In particular, the destylizer features yield only ∼3% accuracy on speaker classification, compared to 20% or higher for HuBERT, Vevo, and CosyVoice 2.0, indicating that most style information has been filtered out. The slightly above-random accuracy of the destylizer may stem from residual correlations between prosody/duration and the destylizer features, as the non-autoregressive design for streaming preserves source utterance duration. Nevertheless, it yields the cleanest content representations among all compared models.

## 5.4 ABLATIONS ON ARCHITECTURE

In this section, we present architectural ablations of StyleStream, all trained on the LMG dataset described in Section 4.1. Specifically, we first evaluate a variant without style embeddings and another that uses FSQ quantized indices instead of pre-quantization continuous features. Results are reported in Table 6. Removing the style embedding significantly reduces S-SIM, A-SIM and E-SIM, showing that relying solely on the unmasked context mel-spectrogram is insufficient for style modeling. The style encoder is therefore essential for faithful style conversion. Moreover, replacing continuous features with FSQ indices severely degrades intelligibility, as shown by the high WER. At first glance, this may seem surprising, since Vevo achieves reasonable results using discrete indices. However, as shown in Table 5, our destylizer's continuous features already carry even less accent, emotion, and speaker information than Vevo's discrete indices. Further quantization strips away critical linguistic content, which explains the poor downstream performance.

Table 6: Ablation studies on the use of style encoder and the choice of destylizer content features.

| Model | WER (%) ↓ | S-SIM ↑ | A-SIM ↑ | E-SIM ↑ | UTMOS ↑ |
|---|---|---|---|---|---|
| StyleStream (offline) | **10.7** | **0.837** | **0.626** | **0.733** | 3.22±.029 |
| w/o Style Emb | 15.3 | 0.775 | 0.509 | 0.653 | **3.47**±.024 |
| w/ FSQ Indices | 123.5 | 0.829 | 0.573 | 0.717 | 2.41±.024 |

Table 7: Ablation on destylizer FSQ bottleneck size. "*" denotes the architecture used in all main experiments.

| FSQ Levels | WER (%) ↓ | S-SIM ↑ | A-SIM ↑ | E-SIM ↑ | UTMOS ↑ |
|---|---|---|---|---|---|
| [7,5,5,5,5] | 13.5 | 0.745 | 0.439 | 0.638 | **3.58**±.022 |
| [5,5,3,3] | 14.9 | 0.822 | 0.600 | 0.725 | 3.17±.031 |
| [5,3,3]* | **10.7** | **0.837** | **0.626** | **0.733** | 3.22±.029 |
| [3,3,3] | 19.0 | 0.825 | 0.618 | 0.732 | 3.13±.032 |
| [5,3] | 101.4 | 0.834 | 0.582 | 0.714 | 2.70±.029 |

We also ablate the effect of the FSQ bottleneck size by varying the FSQ levels during destylizer training and retraining the stylizer on top (Table 7). Enlarging the codebook to $7 \times 5 \times 5 \times 5 \times 5 = 4375$ improves UTMOS, since richer information is kept in the destylizer features, making it easier for the stylizer to reconstruct speech during training. However, this comes at the cost of a sharp drop in A-SIM and E-SIM, indicating substantial style leakage. Conversely, reducing the bottleneck size to $3 \times 3 \times 3 = 27$ degrades the intelligibility, and a further reduction to $5 \times 3 = 15$ severely damages content preservation, indicating an overly restrictive bottleneck. These results suggest that our chosen setting of [5, 3, 3] provides a favorable trade-off: compact enough to filter out most of the style, yet wide enough to preserve linguistic content.

## 6 CONCLUSION

In this work, we presented StyleStream, the first streamable zero-shot voice style conversion system capable of modifying timbre, accent, and emotion in real time, with an end-to-end latency of around 1 second. The framework is built upon two core components: a destylizer, which disentangles linguistic content from style using ASR supervision and a compact FSQ bottleneck, and a stylizer, which leverages a diffusion transformer to reintroduce target style. By adopting continuous pre-quantization features, StyleStream achieves cleaner content–style separation and state-of-the-art voice style conversion performance.

## ETHICAL CONSIDERATIONS

The ethical concerns surrounding StyleStream arise from the broader risks associated with voice cloning and generative speech models, notably the potential for impersonation and privacy violations. Therefore, we will actively work on watermarking and detection models to prevent misuse of this technology.

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

# A  BASELINES

**FACodec (Ju et al., 2024):**  FACodec uses an information bottleneck in combination with a gradient reversal layer and an autoencoding objective to disentangle speech into normalized pitch, phoneme content, and speaker residual features.  Voice conversion is achieved by swapping the speaker embeddings. For evaluation, we use the official checkpoint[8].

**CosyVoice 2.0 (Du et al., 2024b):**  CosyVoice 2.0 introduces a supervised semantic speech tokenizer trained with an ASR loss and finite scalar quantization (FSQ) using a large codebook of 6561 entries. On top of these semantic tokens, a flow-matching transformer is trained with a spectrogram in-painting objective, conditioned on both the semantic tokens and a speaker embedding.  Voice conversion is performed by conditioning on the target speech spectrogram and speaker embedding, together with concatenated target and source semantic tokens. We use the official checkpoint[9] for evaluation.

**SeedVC v2:**  SeedVC v2 is the successor of SeedVC (Liu, 2024), providing support for voice style conversion. As no accompanying paper has been released, the implementation details remain undocumented. We therefore use the official code[10] directly for evaluation.

**Vevo (Zhang et al., 2025b):**  Vevo is the previous state-of-the-art voice style conversion system, featuring self-supervised content tokens with a codebook of 32 codes, autoregressive content-style modeling, and Voicebox-style (Le et al., 2023) acoustic modeling.  We use the official code[11] for evaluation.

**Vevo 1.5 (Li et al., 2025):**  Vevo 1.5 is an upgraded version of Vevo that is designed for not only voice style conversion, but also singing voice conversion. Beyond the original architecture, Vevo 1.5 also introduces a prosody tokenizer that provides coarse-grained prosody tokens to capture melody contours. The official checkpoint[12] is used for evaluation.

# B  SUBJECTIVE EVALUATION

## B.1  BACKGROUNDS OF SUBJECTS

We use Prolific[13] to recruit participants for subjective evaluations. All participants are based in the US or UK, with English as their first language and familiarity with common English accents. Each model receives a total of 400 ratings per test.

## B.2  NATURALNESS MOS

We have developed an automated NMOS evaluation interface, shown in Figure 4. We ask the subjects to rate the naturalness of each speech sample on a scale of 1 to 5, with 1 being completely unnatural and 5 being completely natural.

## B.3  SIMILARITY MOS

We have also developed an automated SMOS evaluation interface. We ask the subjects to rate the similarity of each test sample to the target sample on a scale of 1 to 5, with 1 being not similar at all and 5 being very similar. An example of accent SMOS is shown in Figure 5. We ask the subjects to ignore speaker timbre, emotion or recording quality, but just focus on accent similarity.

---

[8]`https://github.com/open-mmlab/Amphion/tree/main/models/codec/ns3_codec`
[9]`https://github.com/FunAudioLLM/CosyVoice`
[10]`https://github.com/Plachtaa/seed-vc`
[11]`https://github.com/open-mmlab/Amphion/tree/main/models/vc/vevo`
[12]`https://github.com/open-mmlab/Amphion/tree/main/models/svc/vevosing`
[13]`https://www.prolific.com/`

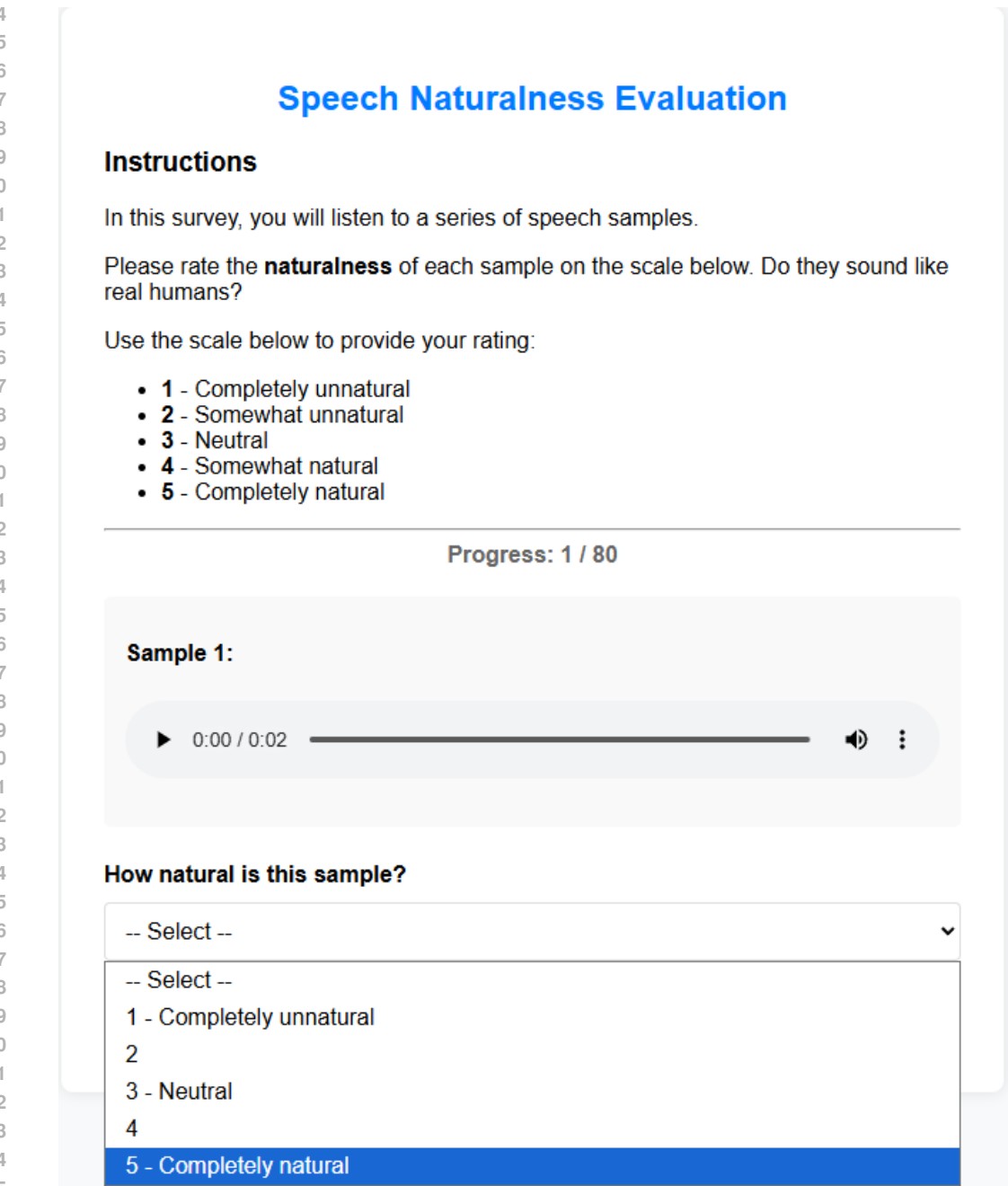

Figure 4: NMOS evaluation interface.

## C  TRAINING HYPERPARAMETERS

Table 8 lists the hyperparameters used to train the accent, emotion, and speaker classifiers with ECAPA-TDNN. The classifiers share the same architecture and settings, with the main difference being the number of classes: 5 for accent, 6 for emotion, and 1251 for speaker classification.

Table 8: ECAPA-TDNN training configurations.

| Hyperparameter | Value |
| --- | --- |
| *Training Configuration* | |
| Initial Learning Rate | 1.0e-6 |
| Weight Decay | 1.0e-5 |
| Gradient Clipping | 1.0 |
| Effective Batch Size | 32 |
| Warmup Steps | 750 |
| Total Training Steps | 8000 |
| Label Smoothing | 0.1 |
| Max Audio Length (s) | 10.0 |
| Linear Neurons | 256 |
| *ECAPA-TDNN Architecture* | |
| Channels | [512, 512, 512, 512, 1536] |
| Kernel Sizes | [5, 3, 3, 3, 1] |
| Dilations | [1, 2, 3, 4, 1] |
| Attention Channels | 128 |
| Res2Net Scale | 8 |
| SE Channels | 128 |
| Dropout | 0.1 |
| Number of Classes | (5, 6, 1251) |

## D  USE OF LARGE LANGUAGE MODELS

We made limited use of large language models (LLMs) during the preparation of this paper. In particular, LLMs were used for grammar checking, rephrasing for clarity, and improving the readability of drafts. In addition, LLMs were employed to generate the female and angry emojis used in Figure 1.

Figure 5: Accent SMOS evaluation interface. The interfaces for the other style SMOS tests follow the same design, with only minor differences in the instructions regarding what aspects to emphasize or disregard.

