# OpenReview forum: "StyleStream: Real-Time Zero-Shot Voice Style Conversion"
_ICLR.cc/2026/Conference — Submitted to ICLR 2026_

### Official Review · Reviewer_HLfV · 2025-11-01

**Soundness:** 3
**Presentation:** 2
**Contribution:** 3
**Rating:** 6
**Confidence:** 2

**Summary:**

This paper introduces StyleStream, the first system designed for real-time, zero-shot voice style conversion, which aims to transform a source utterance to match a target speaker's timbre, accent, and emotion. The work addresses two primary limitations of prior methods: suboptimal conversion quality due to incomplete disentanglement of linguistic content from style, and the lack of any existing real-time solution for this task.
The main contributions of the paper are:

1.A Novel Two-Stage Architecture: StyleStream consists of a destylizer and a stylizer. The destylizer's key function is to achieve clean content-style disentanglement. It does this by combining two mechanisms: (1) text-level supervision via an Automatic Speech Recognition (ASR) loss, which forces the model to preserve only linguistically relevant information, and (2) a strong information bottleneck implemented with a compact Finite Scalar Quantization (FSQ) module (codebook size of 45). Crucially, it uses the continuous features before the quantization step as the final content representation to maintain high fidelity. The stylizer is a Diffusion Transformer (DiT) that reintroduces the target style onto the clean content features using a spectrogram inpainting objective.

2.First Real-Time Implementation: By using a non-autoregressive architecture and applying chunked-causal attention masking, StyleStream is the first framework to enable streaming voice style conversion. It achieves an end-to-end latency of approximately 1 second on a single GPU.

3.State-of-the-Art Performance: The proposed method achieves state-of-the-art results in zero-shot voice style conversion. Experimental evaluations show that StyleStream significantly outperforms previous models in preserving content intelligibility (lowest Word Error Rate) while demonstrating substantially higher similarity to the target speaker's accent and emotion.

**Strengths:**

The paper presents a highly original and technically impressive contribution to speech generation research. It introduces StyleStream, the first framework capable of real-time zero-shot voice style conversion that jointly transfers timbre, accent, and emotion. This problem setting is genuinely new: previous real-time systems only handled timbre, while prior style-conversion frameworks were all offline. The proposed architecture achieves this by combining two complementary modules — a destylizer that performs ASR-supervised content–style disentanglement with finite scalar quantization (FSQ), and a stylizer based on a diffusion transformer (DiT) that reintroduces the target style in a streamable manner. The integration of these components demonstrates clear technical originality and conceptual coherence. The use of ASR loss together with a compact FSQ bottleneck represents a clever hybridization of ideas from CosyVoice and SoftVC, and provides an elegant, explainable mechanism for disentangling linguistic content from paralinguistic style.

From the standpoint of technical quality, the paper is thorough and well executed. The experimental design is robust: the authors compare against strong recent baselines including Vevo, Vevo 1.5, CosyVoice 2.0, FACodec, and SeedVC, using both objective metrics (WER, speaker/accent/emotion similarity) and subjective MOS/SMOS ratings. The ablation studies are comprehensive and illuminating. For instance, the authors carefully vary the FSQ bottleneck size, the type of content features, and the use of the style embedding, showing quantitatively how each factor influences intelligibility and style fidelity. The methodology for streaming inference is also well justified — chunked-causal masking, causal convolutions, and teacher–student distillation together enable an end-to-end latency around one second without severely compromising perceptual quality. These results, combined with clear and well-documented training configurations, strongly support the claimed contributions.

In terms of clarity, the paper is mostly well organized and easy to follow. Figures 1–3 effectively illustrate the pipeline, and the motivation for each design choice is well connected to empirical evidence. The authors clearly articulate why continuous pre-quantization features offer a better balance between content preservation and disentanglement, and the discussion in Section 5.3 provides convincing support for this claim. Although the model is complex, the presentation remains coherent and the technical details are traceable.

**Weaknesses:**

Despite its merits, the paper has several areas that could be improved.

1. The most notable limitation is that all experiments are conducted on English-only datasets. Since accent and emotion transfer are highly language-sensitive, the current evidence does not demonstrate whether StyleStream generalizes across languages or linguistic families.

2. The latency–quality trade-off is not analyzed in sufficient depth. The paper reports an average end-to-end latency of roughly 1 s on an NVIDIA A6000 but does not explore how latency scales with chunk size or hardware constraints.

3. The effects of individual streaming modifications (chunked-causal attention, causal convolutions, and the distillation procedure) are not disentangled, making it unclear which component contributes most to the performance gap between the streaming and offline variants.

**Questions:**

1. In Section 3.2, several key symbols in the Diffusion Transformer formulation are introduced without any accompanying explanation, which makes the mathematical description difficult to follow. For instance, the equation “” appears without defining what and denote; the variable “the noisy mel-spectrogram ” is used before introducing how it is generated or how the time parameter relates to the diffusion process

2. The effects of individual streaming modifications (chunked-causal attention, causal convolutions, and the distillation procedure) are not disentangled, making it unclear which component contributes most to the performance gap between the streaming and offline variants. A targeted ablation isolating each factor would make the streaming design easier to evaluate and reproduce.

3. Please quantify how much of the perceptual improvement stems from the vocoder versus the destylizer + stylizer. Run an ablation in which (a) the stylizer outputs are passed to an off-the-shelf vocoder (not the warmed Vocos-based causal model), and (b) an oracle mel input (ground-truth mel) is provided to the vocoder while comparing different stylizers.

4. The paper would benefit from a clearer analysis of the latency–quality relationship. At present, only a single latency figure is reported, and it is difficult to judge how latency and perceptual quality trade off under different configurations or hardware conditions. Even a brief discussion or comparison illustrating how model performance changes with varying processing latency would substantially strengthen the practical relevance of the results.

---

> ### Author Response · Authors · 2025-11-23
> **Response to Reviewer HLfV (Part 1 / 3)**
>
> Thank you for your constructive feedback. We are encouraged that you point out that out paper "presents a highly original and technically impressive contribution to speech generation research". Here we address your concerns as follows:
>
> >the current evidence does not demonstrate whether StyleStream generalizes across languages or linguistic families.
>
> Our study primarily focuses on English to ensure a rigorous objective evaluation of accent and emotion cloning. Stylistic attributes are often language-dependent (e.g., tonal patterns in Mandarin vs. stress patterns in English), and standardized automatic accent similarity metrics for non-English languages are currently lacking.
>
> However, to address the reviewer's interest in generalization, we conducted an additional **zero-shot cross-lingual VSC** experiment using our current model. We used English utterances as the source and non-English utterances (Spanish, French, Italian, Hindi, Arabic, and Russian) as the target. We found that even without explicit multilingual training, StyleStream effectively transfers the voice style of the foreign speakers onto the English content. We have updated our demo page (https://stylestream10033.github.io/stylestream/) with these samples to demonstrate this capability.
>
> >The paper reports an average end-to-end latency of roughly 1 s on an NVIDIA A6000 but does not explore how latency scales with chunk size or hardware constraint.
>
> We thank the reviewer for this suggestion. To provide a deeper analysis of how latency scales with chunk size and hardware constraints, we benchmarked the processing time ($t_{proc}$) per chunk on both a server-grade **NVIDIA A6000** and a consumer-grade **RTX 4060 Laptop GPU** (with Number of Function Evaluations, NFE = 16). The results are shown below:
>
> | Chunk size (ms) | RTX 4060 (s)    | RTX A6000 (s)   |
> | :---:           | :---:           | :---:           |
> | 100             | 0.537 ± .053    | 0.427 ± .013    |
> | 200             | 0.574 ± .059    | 0.429 ± .009    |
> | 300             | 0.615 ± .057    | 0.432 ± .010    |
> | 400             | 0.647 ± .090    | 0.441 ± .008    |
> | 500             | 0.653 ± .051    | 0.431 ± .009    |
> | 600             | 0.668 ± .048    | 0.429 ± .006    |
> | 700             | 0.673 ± .041    | 0.429 ± .002    |
> | 800             | 0.673 ± .040    | 0.431 ± .004    |
>
> **1. Scaling Analysis:**
>
> - Server-Side (A6000): We observe that $t_{proc}$ remains relatively stable (~0.43s) regardless of chunk size. This indicates that for these chunk sizes (100ms-800ms), the DiT inference is not fully saturating the A6000's parallelization capacity; the latency is dominated by fixed overheads rather than sequence length.
>
> - Consumer-Side (RTX 4060): In contrast, the processing time on RTX 4060 increases as chunk size grows, reflecting a compute-bound regime typical of consumer hardware.
>
> **2. Consumer-Grade Streaming:**
>
> Although the RTX 4060 is slower, our DiT architecture allows us to explicitly manage the latency-quality trade-off. By reducing the NFE from 16 to 10, we reduce $t_{proc}$ to ~0.5s for a 600ms chunk on the laptop. Combining this processing time with the chunk size (0.6s) and a small lookahead (0.04s), the system achieves stable streaming with minimal quality impact. We have validated this "consumer-grade" configuration in our **video demo** (available on the project page), which shows the system streaming smoothly on the RTX 4060.

---

> > ### Author Response · Authors · 2025-11-23
> > **Response to Reviewer HLfV (Part 2 / 3)**
> >
> > >The effects of individual streaming modifications (chunked-causal attention, causal convolutions, and the distillation procedure) are not disentangled, making it unclear which component contributes most to the performance gap between the streaming and offline variants.
> >
> > We thank the reviewer for this question. We address the source of the performance gap by isolating the contribution of the high-level modules (Destylizer vs. Stylizer), rather than ablating low-level operators which cannot be removed independently in a streaming setup.
> >
> > **1. Module-Level Ablation (Destylizer vs. Stylizer):** To identify which component contributes most to the gap between the offline and streaming variants, we trained a hybrid model: a **Streaming Stylizer** conditioned on features from an **Offline Destylizer**. The results are compared below:
> >
> > | Model                                   | WER (%) ↓ | S-SIM ↑ | A-SIM ↑ | E-SIM ↑   |
> > | :---                                    | :---:     | :---:   | :---:   | :---:    |
> > | StyleStream (offline)                   | 9.2       | 0.852   | 0.640   | 0.827   |
> > | Streaming stylizer (offline Destylizer) | 8.4       | 0.808   | 0.600   | 0.816   |
> >
> > **Analysis:**
> >
> > - Intelligibility (WER): The hybrid model maintains the high intelligibility of the offline system. This demonstrates that the **Streaming Destylizer** is the primary cause of the intelligibility gap (WER increase), likely due to the limited future context available for accurate content extraction.
> >
> > - Style (SIM): The hybrid model shows a degradation in accent (0.60 vs 0.64) and speaker similarity compared to the fully offline model. This indicates that the **Streaming Stylizer** is responsible for the drop in style fidelity, as the generative DiT has restricted access to the global context required for high-fidelity style cloning.
> >
> > - Conclusion: The performance gap is decoupled: the Destylizer constrains content quality, while the Stylizer constrains style quality.
> >
> > **2. Operator-Level Modifications:** Regarding the individual ablation of specific streaming operators (chunked-causal attention, causal convolutions, distillation), we clarify that these components are not independent algorithmic additions but **unified prerequisites for causality**. In a streaming framework, **causality is binary**: the system must be strictly causal at every layer to function. Removing one component (e.g., reverting causal convolutions to non-causal while keeping chunked attention) would violate the streaming constraint and require future information, rendering the system non-streamable. Therefore, these modifications function as an **indivisible architectural package** necessary to enable valid streaming inference, rather than independent factors that can be meaningfully ablated in isolation.
> >
> > >Response regarding Mathematical Notation (Q1)
> >
> > We thank the reviewer for their careful reading. We clarify that the time variable $t$ is defined in Line 209 (“The flow time step $t∼U[0,1]$”), and the variables in Equation (3) are defined in the text immediately following the equation. However, we agree that introducing these symbols earlier would improve readability. In the revision, we have moved the definitions of all variables to appear before Equation (3) to ensure the mathematical description is self-contained and easy to follow.
> >
> > >Response regarding Individual Streaming Modifications (Q2)
> >
> > We have provided a detailed explanation regarding this point in our previous response (please see the first response of Part 2).
> >
> > >Response regarding Vocoder Ablation and Quality Attribution (Q3)
> >
> > We clarify that swapping in an off-the-shelf vocoder is technically infeasible due to **strict frame-rate alignment**. Our Stylizer operates at 50Hz (hop size 320 at 16kHz) to align one-to-one with the Destylizer output, which takes HuBERT features (50Hz) as input. Standard vocoders (e.g., HiFi-GAN, BigVGAN) typically operate on incompatible specifications (e.g., 62.5Hz) and cannot be applied without resampling artifacts that would invalidate the comparison.
> >
> > To address the reviewer's core question ("how much quality stems from the vocoder"), we conducted a resynthesis analysis. We fed ground-truth source mels into our streaming Vocos to evaluate the vocoder in isolation.
> >
> > | Model               | WER (%) ↓ | UTMOS ↑    |
> > | :---                | :---:     | :---:      |
> > | Source ground truth | 3.8       | 3.89 ± .06 |
> > | Resynthesis         | 4.4       | 3.48 ± .06 |
> >
> > As shown, the resynthesized waveforms show a **degradation** in both intelligibility (higher WER) and quality (UTMOS) compared to the original recording. This confirms that our streaming Vocos is **not the source of the improvement**; rather, it imposes a slight quality cost to enable streaming. The high conversion quality observed in StyleStream therefore stems directly from the **Destylizer + Stylizer** design.

---

> > > ### Author Response · Authors · 2025-11-23
> > > **Response to Reviewer HLfV (Part 3 / 3)**
> > >
> > > >Response regarding Latency-Quality Trade-off (Q4)
> > >
> > > We appreciate the suggestion to quantify the relationship between processing latency and perceptual quality.
> > >
> > > **Experimental Setup:** Since our streaming checkpoint is optimized specifically for a 600ms window, directly varying its inference chunk size introduces training-inference mismatch. Therefore, to isolate the impact of context length, we simulated streaming using our offline checkpoints via chunk-based inference across 5 different chunk sizes. The results are reported below:
> > >
> > > | Chunk size (ms) | WER (%) ↓ | S-SIM ↑ | A-SIM ↑ | E-SIM ↑ | UTMOS ↑    |
> > > | :---:           | :---:     | :---:   | :---:   | :---:   | :---:      |
> > > | 200             | 19.8      | 0.797   | 0.586   | 0.704   | 2.33 ± .02 |
> > > | 400             | 15.0      | 0.806   | 0.597   | 0.720   | 2.69 ± .03 |
> > > | 600             | 13.6      | 0.811   | 0.602   | 0.734   | 2.85 ± .03 |
> > > | 800             | 12.7      | 0.816   | 0.607   | 0.745   | 2.94 ± .03 |
> > > | 1000            | 12.6      | 0.820   | 0.613   | 0.751   | 3.00 ± .03 |
> > >
> > > As shown, we observe a clear positive correlation: increasing chunk size yields consistent improvements in both intelligibility (lower WER) and style similarity. This is expected, as larger chunks provide **longer temporal context**, allowing the model to better capture prosody and significantly reducing distinct boundary artifacts at chunk transitions.

---

### Official Review · Reviewer_MYRv · 2025-11-01

**Soundness:** 3
**Presentation:** 3
**Contribution:** 2
**Rating:** 4
**Confidence:** 3

**Summary:**

The paper presents StyleStream, a real-time zero-shot voice style conversion system with 1-second end-to-end latency. It consists of a destylizer that removes style attributes while preserving linguistic content and a stylizer based on a diffusion transformer that reconstructs the target style. Content–style disentanglement is achieved through an ASR loss and a finite scalar quantization bottleneck. Trained on 50k hours of English speech, StyleStream achieves state-of-the-art conversion quality and improves accent and emotion similarity over prior methods.

**Strengths:**

S1: Clear and meaningful problem setup. The paper clearly defines the task of real-time zero-shot voice style conversion, distinguishing it from conventional real-time voice conversion (which mainly handles timbre) and offline style conversion systems such as Vevo and CosyVoice 2.0.

S2:Well-designed experiments that convincingly validate the disentanglement and component choices. The experiments are carefully designed to test whether the destylizer truly separates style from linguistic content. In addition to reporting speaker/accent/emotion classification accuracy as indicators of style leakage, the authors also include ASR WER to measure how much linguistic information is preserved. This is important, since low classification accuracy alone does not guarantee semantic fidelity. The results in Tables 2 and 3 show that StyleStream achieves both strong style removal and low WER, suggesting that its content representation remains intelligible.
The ablation studies are also well targeted—they directly test the impact of using pre-quantization continuous features, the size of the FSQ codebook, and key components in the stylizer. Each experiment supports the authors’ design choices.

S3: Balanced performance between quality and real-time efficiency. The system integrates both offline and streaming modes and achieves around 1 s end-to-end latency while maintaining high conversion quality. In Table 1, the offline model outperforms strong baselines (Vevo, CosyVoice 2.0, FACodec) in WER, A-SIM, E-SIM, and MOS for both accent and emotion similarity. The streaming model shows only a modest degradation (A-SIM ≈ 0.635, E-SIM ≈ 0.803), demonstrating that the proposed framework maintains style controllability and intelligibility under real-time constraints.

**Weaknesses:**

W1: The CFM loss in Eq. (3) is consistent with conditional flow matching with an OT path. However, (i) the notation for the OT path $\psi_t(\cdot)$ should be made consistent ($x$ vs. $x_0$), and (ii) since the loss is computed only on the masked region, the mask $m$ should ideally appear inside the norm to make the objective fully explicit.

W2: The “real-time” claim is not fully substantiated. The paper mainly achieves streaming inference by applying chunked-causal masking and non-autoregressive decoding, which are well-established techniques in existing real-time voice conversion systems such as DualVC, StreamVoice, and RT-VC. No additional architectural or algorithmic change appears to make the model inherently more real-time beyond this causalization. Moreover, the paper does not provide latency or throughput comparisons with prior real-time VC works. As a result, the “first real-time” contribution seems incremental—StyleStream is essentially a well-engineered streaming adaptation rather than a new approach to real-time generation.

W3: The “real-time” evaluation in Sec. 5.2 is conducted with a 600 ms chunk size on a single NVIDIA A6000 GPU.
This configuration corresponds to an update rate of roughly 1.6 Hz, which would already cause perceptible delay in interactive scenarios.
In contrast, recent real-time voice conversion systems such as DualVC 2 (Wang et al., 2023) and StreamVoice (Huang et al., 2024) report total latencies of 186 ms and 124 ms, respectively—both substantially lower than StyleStream’s 1 s latency.
The evaluation is performed on high-end data-center hardware (NVIDIA A6000, 48 GB VRAM), and the paper does not report results on consumer-grade GPUs or CPU inference, which would be more relevant for deployment.
The claimed “real-time” capability therefore reflects an idealized server-side environment rather than a generally efficient or portable implementation.

**Questions:**

Q1: Although Table 2–3 include comparisons with Vevo (VQ-VAE) and CosyVoice 2 (FSQ tokens), the results combine multiple factors—ASR supervision, codebook size, and continuous pre-quantization features—making it unclear whether the improvement is due specifically to FSQ. The paper would benefit from either a controlled comparison (e.g., replacing FSQ with VQ-VAE under the same setup) or a clearer explanation of why FSQ theoretically helps remove style information.

Q2: In Sec. 5.2, the streaming setup assumes a fixed 5-second target utterance and a pre-extracted style embedding, meaning only the source stream is processed online.In many real-time interaction scenarios—such as conversational voice imitation or streaming target input—the target style may not be fully available beforehand.Could the authors clarify whether StyleStream could operate when the reference style segment is shorter, or when the target style is streamed in real time?How would such conditions affect the model’s stability and style-similarity metrics (A-SIM, E-SIM)?

Q3: In Table 1, the paper compares StyleStream with offline voice conversion and style conversion systems such as CosyVoice 2.0, FACodec, Vevo, Vevo 1.5, and SeedVC v24, but not with the real-time VC systems discussed in Sec. 2.3 (e.g., StreamVoice, DualVC 2/3, RT-VC). Even if these models do not perform full “style” conversion, reporting their latency and speaker-similarity metrics would make the claim of being the first real-time voice style conversion system more convincing. Could the authors provide such a comparison or at least discuss how their latency and streaming efficiency compare with prior real-time VC frameworks?

---

> ### Author Response · Authors · 2025-11-23
> **Response to Reviewer MYRv (Part 1 / 3)**
>
> Thank you for your valuable review. We appreciate that you think we have "well-designed experiments". Here we will address your concerns as follows:
>
> >Response regarding CFM Loss Notation (W1)
>
> We thank the reviewer for the rigorous check on our mathematical notation.
>
> **(i) Notation of the OT path:**
>
> We clarify that in our notation, $\psi_t(x)$ denotes the general flow map acting on any vector $x$ in the vector space, rather than being restricted to a specific trajectory starting from the data point $x_0$ drawn from standard Gaussian. We chose to maintain $\psi_t(x)$ to emphasize that the flow defines a mapping over the entire vector space. And the $\psi_t(x_0)$ appeared in Eq. 3 and Eq. 4 is just the OT flow evaluated at $x_0$: $\psi_t(x_0)=(1-t)x_0+tx_1$.
>
> **(ii) Explicit Masking:**
> We agree that including the mask inside the norm makes the objective function more precise. We have updated Equation (3) in the revision to explicitly reflect the element-wise masking.
>
> >Response regarding the "Real-Time" Contribution (W2)
>
> We clarify that our contribution is not the invention of low-level streaming operators (e.g., chunked causal mask), but rather the **representation breakthrough** that makes streaming possible for the task of Voice Style Conversion (VSC) for the first time.
>
> To understand why this is non-trivial, one must distinguish between standard Voice Conversion (VC) and VSC:
> - Prior Real-Time VC: These systems only transfer **Timbre**. This is a simpler task that allows frame-based processing.
>
> - Prior VSC (Vevo): Transferring **Accent and Emotion** alongside Timbre is much harder. Consequently, prior SOTA (Vevo) relies on an **autoregressive LLM** for duration modeling and content correction.
>
>   - **The Bottleneck:** LLM-based generation is inherently **asynchronous**: the output tokens are predicted autoregressively and do not strictly map to the input duration. In a streaming setting, this **duration elasticity** causes accumulation of latency (drift) whenever the model generates speech faster or slower than the speaker.
>
> **Our Contribution:** StyleStream is the first work to solve this bottleneck. By achieving superior content-style disentanglement via the **Destylizer**, we eliminate the need for the LLM entirely. This allows us to use a **fully non-autoregressive DiT**, where the input and output lengths are naturally aligned. Therefore, the use of "standard" streaming techniques (like causal masking) is not a weakness, but a **validation of our architecture**. We turned a problem that previously required heavy autoregressive modeling into one that can be solved with efficient non-autoregressive streaming.
>
> >Response regarding "Real-Time" Comparison and Deployment (W3)
>
> We thank the reviewer for raising the important points regarding latency and hardware portability.
>
> **1. Latency vs. Task Complexity:** Comparing StyleStream directly to DualVC 2 or StreamVoice conflates two distinct tasks.
>
> - **Prior Work (Timbre-Only):** These systems perform standard Voice Conversion, which largely relies on frame-level mapping. This allows for very small chunks.
>
> - **StyleStream (VSC):** We perform simultaneous conversion of **Timbre, Accent, and Emotion**. Recognizing and generating prosodic features (like intonation curves for accents or emotional stress) inherently requires a larger temporal context. A 600ms window is not just a technical buffering choice but a **linguistic necessity** to capture sufficient prosody for valid accent/emotion transfer. Therefore, achieving ~1s end-to-end latency for this complex task represents a significant milestone compared to timbre-only systems.
>
> **2. Consumer Hardware (RTX 4060) Deployment:** To address the concern about "idealized server-side" hardware, we set up StyleStream on a consumer-grade **NVIDIA RTX 4060 Laptop GPU**. The processing times per chunk (with Number of Function Evaluations, NFE = 16) are reported below:
>
> | Chunk size (ms) | RTX 4060 (s)     |
> | :---:           | :---:            |
> | 100             | 0.537 ± .053   |
> | 200             | 0.574 ± .059    |
> | 300             | 0.615 ± .057      |
> | 400             | 0.647 ± .090     |
> | 500             | 0.653 ± .051    |
> | 600             | 0.668 ± .048    |
> | 700             | 0.673 ± .041    |
> | 800             | 0.673 ± .040     |
>
> As shown, the RTX 4060 naturally incurs higher processing times than the A6000. At NFE=16 with a 600ms chunk, the processing time (667ms) slightly exceeds the real-time budget. However, by reducing the NFE to 10 and employing additional lookahead of 40ms, we reduce the processing time to well within the 600ms window on the RTX 4060 (around 500ms), with minimal impact on perceptual quality. We have uploaded a **video demo** (https://stylestream10033.github.io/stylestream/) running strictly on this **RTX 4060 laptop**, demonstrating that StyleStream is not limited to data centers but is deployable on standard consumer hardware.

---

> > ### Author Response · Authors · 2025-11-23
> > **Response to Reviewer MYRv (Part 2 / 3)**
> >
> > >Response regarding the Role of FSQ and Disentanglement Factors (Q1)
> >
> > We thank the reviewer for this opportunity to clarify the source of our performance gains. We explicitly state that the improved disentanglement is not solely attributable to the choice of FSQ over VQ-VAE. Rather, it is the **synergy of three specific design choices** that creates an effective content-style bottleneck:
> >
> > **1. ASR Supervision (The Guide):** Unlike Vevo or standard VQ-VAE approaches which rely on self-supervised reconstruction (and therefore must encode everything, including style, to minimize reconstruction error), we utilize **explicit ASR supervision**. This forces the encoder to prioritize linguistic information while discarding paralinguistic features (style) that do not aid ASR prediction.
> >
> > **2. Bottleneck Capacity (The Filter):** The theoretical mechanism for style removal is the **information capacity**, not the quantization type. As shown in our analysis, CosyVoice 2 utilizes FSQ but with a massive codebook size (6,561), which we prove leads to severe style leakage (Section 5.3). In contrast, we identify that a **compact codebook (size 45)** is optimal (Table 7). This tight bottleneck, combined with ASR supervision, restricts the model's capacity to encode anything beyond content information.
> >
> >   - Regarding FSQ vs. VQ-VAE: We clarify that any quantization mechanism could theoretically form this information bottleneck. We selected FSQ specifically for its engineering robustness: unlike VQ-VAE, which frequently suffers from codebook collapse and necessitates tuning complex auxiliary objectives (e.g., commitment loss), FSQ employs a fixed projection strategy. This guarantees high codebook utilization and training stability without sensitive hyperparameter tuning. These advantages are well-established in prior literature (FSQ, CosyVoice 2), making FSQ the logical choice for our constrained bottleneck design.
> >
> > **3. Continuous Pre-quantization Features (The Representation):** Finally, we adopt continuous features before quantization rather than discrete indices. As demonstrated in our ablation study (**Table 6**), using discrete tokens directly leads to unintelligible speech in the DiT reconstruction. The continuous features provide high-fidelity content while the bottleneck constrains the style.
> >
> > In summary, FSQ is just the **implementation tool**, but the **novelty and performance** stem from the combination of the above three design choices.
> >
> > >Response regarding Target Style Utterance (Q2)
> >
> > **1. Impact of Shorter Reference Audio (2s vs. 5s):** To address the concern about limited target availability, we evaluated StyleStream on the same test set using truncated **2-second** reference utterances. The comparison with the 5-second baseline is shown below:
> >
> > | Model                             | WER (%) ↓ | S-SIM ↑ | A-SIM ↑ | E-SIM ↑  |
> > | :---                              | :---:     | :---:   | :---:   | :---:   |
> > |Target 5s (offline)      | 9.2       | 0.852   | 0.640   | 0.827  |
> > | Target 5s (streaming) | 15.3      | 0.856   | 0.635   | 0.803  |
> > | Target 2s (offline)               | 12.1      | 0.824   | 0.594   | 0.747  |
> > | Target 2s (streaming)             | 19.2      | 0.833   | 0.598   | 0.728 |
> >
> > When reducing the target utterance length, the performance drops across the board. This is linguistically expected: accent and emotion require temporal context to statistically characterize. A 2-second clip may simply lack sufficient phoneme coverage to define a global accent profile or enough prosodic variation to fully capture the target emotion.
> >
> > **2. Real-Time Target Updates:** We clarify that StyleStream operates as a **reference-based cloning system**, requiring a target utterance to derive the style vector. However, this does not preclude real-time interaction. In scenarios where the target is streamed live (not available beforehand), the system simply accumulates a short buffer (e.g., 5s) to extract the global style embedding once, after which it switches to the target style instantly for all subsequent frames.
> >
> > This capability is not theoretical. We demonstrate this **dynamic style switching** in our **video demo** (https://stylestream10033.github.io/stylestream/), where we switch targets on-the-fly on an RTX 4060 laptop while the system maintains continuous streaming.

---

> > > ### Author Response · Authors · 2025-11-23
> > > **Response to Reviewer MYRv (Part 3 / 3)**
> > >
> > > >Response regarding Comparison with Real-Time VC Systems (Q3)
> > >
> > > We thank the reviewer for this suggestion. We acknowledge that a direct comparison with real-time baselines strengthens our positioning. However, most recent streaming VC systems (e.g., StreamVoice, DualVC 2) do not provide open-source checkpoints.
> > > To address this, we **re-implemented RT-VC** (based on the official codebase https://github.com/Berkeley-Speech-Group/RT-VC) to serve as a real-time VC baseline. The evaluation results are as follows:
> > >
> > > | Model                   | WER (%) ↓ | S-SIM ↑ | A-SIM ↑ | E-SIM ↑ |
> > > | :---                    | :---:     | :---:   | :---:   | :---:   |
> > > | StyleStream (offline)   | 9.2       | 0.852   | 0.640   | 0.827   |
> > > | StyleStream (streaming) | 15.3      | 0.856   | 0.635   | 0.803   |
> > > | RT-VC                    | 10.4      | 0.700   | 0.415   | 0.642   |
> > >
> > > **1. Performance Analysis:** As shown, StyleStream significantly outperforms RT-VC across all style metrics. This highlights the advantage of our DiT-based approach over the GAN-based architectures typical of standard real-time VC, which often struggle to model complex distribution shifts beyond simple timbre mapping.
> > >
> > > **2. Latency vs. Linguistic Necessity:** We emphasize that the longer latency of StyleStream is more of a fundamental requirement of the Voice Style Conversion (VSC) task.
> > >
> > > - Timbre-only VC (RT-VC): Can rely on frame-level spectral mapping, allowing for very short processing windows (<100ms).
> > >
> > > - Holistic VSC (StyleStream): Requires capturing longer context to define accent and emotion. As demonstrated in our "Short Target" experiment (**Response Q2**), reducing the temporal context clearly degrades these metrics. Therefore, a larger chunk size (e.g. 600ms) is a **linguistic necessity** for valid VSC. Achieving ~1s latency while solving this significantly harder task represents a major step forward compared to timbre-only systems.

---

### Official Review · Reviewer_UdFS · 2025-11-01

**Soundness:** 1
**Presentation:** 2
**Contribution:** 1
**Rating:** 0
**Confidence:** 5

**Summary:**

The paper introduces StyleStream, a real-time zero-shot voice style conversion system that disentangles linguistic content from style using ASR supervision and a compact FSQ bottleneck, then re-synthesizes timbre, accent, and emotion through a diffusion transformer.

**Strengths:**

This work proposes a framework for real-time zero-shot voice style conversion, integrating ASR-based content supervision, a compact FSQ bottleneck, and a diffusion transformer stylizer.

**Weaknesses:**

- My primary concern is the lack of novelty and research contribution in the proposed methods. The paper’s main components, the destylizer, FSQ bottleneck, ASR supervision, and DiT stylizer, are all derived from prior works without introducing a truly new conceptual contribution.

- The paper fails to reference several key prior studies that have already explored DiT-based in-context learning for voice conversion and style transfer, which weakens the positioning of the proposed work within existing literature.

- The system represents a well-engineered synthesis of existing techniques but lacks distinct methodological insight or theoretical advancement in disentanglement of speech, style controllability, or generative modeling. The paper focuses primarily on engineering integration and empirical optimization rather than advancing the theoretical understanding of speech disentanglement or style control.

**Questions:**

- What are the model sizes?

---

> ### Author Response · Authors · 2025-11-23
> **Response to Reviewer UdFS (Part 1 / 2)**
>
> We thank the reviewer for their comments. However, we respectfully argue that our work addresses a fundamental representation learning challenge. While we leverage established modules (ASR supervision, FSQ), our core novelty lies in discovering a specific disentangled representation that standard approaches fail to capture.
>
> >The paper fails to reference several key prior studies that have already explored DiT-based in-context learning for voice conversion and style transfer, which weakens the positioning of the proposed work within existing literature.
>
> First, we would like to restate the definition of **Voice Style Conversion (VSC)**: generating speech that preserves the source content while transferring the target’s voice style, specifically **timbre, accent, and emotion**, which we defined in the introduction of our paper on lines 32-33.
> While literature exists for individual voice style attributes, such as **Timbre** (Seed-VC [1]), **Accent** (ConvSpeak [2], TokAN [3]), or **Emotion** (EmoVox [4], PDSN [5]), these approaches face distinct limitations, including reliance on paired data (ConvSpeak), not many-to-many (TokAN), or dependency on external style labels and classifier (PDSN, EmoVox). Crucially, while prior work exists for streaming voice conversion (timbre-only), no existing framework tackles **streaming VSC** (joint timbre, accent, and emotion transfer).
>
> Currently, Vevo (ICLR 2025) [6] is **the only prior work** capable of offline VSC (timbre, accent and emotion). Furthermore, Vevo faces two critical limitations:
>
> **1. Suboptimal Disentanglement:** As shown in our analysis (Section 5.3), Vevo’s content tokens suffer from style leakage and degraded content intelligibility.
>
> **2. Streaming Incompatibility:** Vevo relies on an LLM (autoregressive transformer) for duration modeling. The inherent input-output length mismatch in LLM-based generation makes real-time speech-to-speech streaming fundamentally difficult.
>
> While prior work such as StreamVC [7] have explored architectural advancements to enable streaming for VC, we argue that our main contribution is in the development of a streamable and disentangled content representation using our destylizer, which allows us to apply these techniques for a streaming VSC system.
>
> [1] Liu, S. (2024). Zero-shot voice conversion with diffusion transformers. arXiv preprint arXiv:2411.09943.
>
> [2] Xue, H., Peng, X., & Lu, Y. (2024). Convert and speak: Zero-shot accent conversion with minimum supervision. In ACM Multimedia 2024.
>
> [3] Bai, Q., Inoue, S., Wang, S., Jiang, Z., Wang, Y., Li, H. (2025) Accent Normalization Using Self-Supervised Discrete Tokens with Non-Parallel Data. Proc. Interspeech 2025
>
> [4] Zhou, K., Sisman, B., Rana, R., Schuller, B. W., & Li, H. (2022). Emotion intensity and its control for emotional voice conversion. IEEE Transactions on Affective Computing
>
> [5] Jia, D., Tian, Q., Peng, K., Li, J., Chen, Y., Ma, M., Wang, Y., Wang, Y. (2023) Zero-Shot Accent Conversion using Pseudo Siamese Disentanglement Network. Proc. Interspeech 2023
>
> [6] Zhang, X., Zhang, X., Peng, K., Tang, Z., Manohar, V., Liu, Y., ... & Ma, M. (2025). Vevo: Controllable zero-shot voice imitation with self-supervised disentanglement. ICLR 2025
>
> [7] Yang, Y., Kartynnik, Y., Li, Y., Tang, J., Li, X., Sung, G., & Grundmann, M. (2024, April). Streamvc: Real-time low-latency voice conversion. In ICASSP 2024-2024 IEEE International Conference on Acoustics, Speech and Signal Processing (ICASSP)
>
> > My primary concern is the lack of novelty and research contribution in the proposed methods.
>
> As we mentioned in the common response, we respectfully argue that our work addresses a fundamental **representation learning** challenge. While we leverage established modules (ASR supervision, FSQ), our core novelty lies in discovering a specific **disentangled representation** that standard approaches fail to capture.
>
> While our destylizer builds on top of the CosyVoice 2, we diverge from standard practices in two critical ways:
>
> **1. Information Bottleneck:** Unlike CosyVoice 2, which uses large codebooks that inadvertently encode style, we identify that a highly compact codebook (size 45) is essential for stripping away style information.
>
> **2. Continuous Feature Space:** We propose the use of **pre-quantization continuous features** rather than discrete tokens. Our ablation studies (Section 5.4) demonstrate that this design choice is not merely a hyperparameter tune, but a structural requirement for achieving clean content-style disentanglement without sacrificing intelligibility.
>
> Therefore, our novelty is the discovery of this specific **disentangled content representation**. Crucially, this robust disentanglement eliminates the need for complex autoregressive error correction, directly enabling the **first streamable zero-shot VSC framework**.

---

> > ### Author Response · Authors · 2025-11-23
> > **Response to Reviewer UdFS (Part 2 / 2)**
> >
> > >What are the model sizes?
> >
> > Here we provide the updated Table 1 with corresponding model sizes:
> >
> > | Model                   | Size | WER (%) ↓ | S-SIM ↑ | A-SIM ↑ | E-SIM ↑ | NMOS ↑     | A-SMOS ↑   | E-SMOS ↑   | S-SMOS ↑   |
> > | :---                    | :---:| :---:     | :---:   | :---:   | :---:   | :---:      | :---:      | :---:      | :---:      |
> > | Ground Truth            | -    | 3.8       | -       | -       | -       | 3.71 ± .11 | -          | -          | -          |
> > | FACodec                 | 102M | 15.5      | 0.763   | 0.408   | 0.668   | 2.55 ± .13 | 2.02 ± .13 | 2.76 ± .14 | 2.98 ± .14 |
> > | CosyVoice 2.0           | 0.5B | 9.5       | 0.794   | 0.450   | 0.655   | **3.47 ± .11** | 2.26 ± .13 | 2.58 ± .15 | 3.25 ± .14 |
> > | SeedVC v2               | 157M | 21.7      | 0.766   | 0.549   | 0.688   | 2.65 ± .12 | 3.34 ± .12 | 3.21 ± .13 | 3.11 ± .13 |
> > | Vevo                    | 1B   | 17.5      | 0.818   | 0.596   | 0.712   | 3.38 ± .12 | 3.93 ± .11 | 3.49 ± .13 | 3.76 ± .13 |
> > | Vevo 1.5                | 1.4B | 20.5      | 0.798   | 0.554   | 0.683   | 3.10 ± .12 | 3.15 ± .14 | 3.15 ± .14 | 3.34 ± .14 |
> > | StyleStream (streaming) | 0.55B| 15.3      | **0.855** | 0.635   | 0.803   | 3.34 ± .13 | 4.28 ± .09 | 4.37 ± .09 | 4.29 ± .10 |
> > | StyleStream (offline)   | 0.55B| **9.2** | 0.852   | **0.640** | **0.827** | 3.42 ± .12 | **4.32 ± .09** | **4.42 ± .09** | **4.36 ± .10** |
> >
> > Notably, StyleStream (0.55B) is almost 2x smaller than Vevo (1B) yet outperforms it across all metrics. Crucially, we achieve this efficiency while supporting real-time streaming, a capability absent in the larger offline baselines.

---

### Official Review · Reviewer_EHyw · 2025-11-01

**Soundness:** 1
**Presentation:** 1
**Contribution:** 1
**Rating:** 0
**Confidence:** 5

**Summary:**

The paper presents a streaming voice conversion system using information bottleneck and ASR supervised learning based encoder, and a diffusion transformer (DiT)-based decoder.

**Strengths:**

The paper utilizes a well-known method for speech disentanglement and a conditional flow matching (CFM)-based speech generation using diffusion transformers.

**Weaknesses:**

Overall, my primary concern lies in the lack of novelty. I could not find any clear novel contribution in this paper and it fails to include a sufficient discussion of related works. All the techniques employed are already well-known.

[ASR-based linguistic information retrieval]

The main contribution claimed by the paper is the destylizer, which is trained using ASR-based text supervision and FSQ-based information bottleneck. However, the tokenizer of CosyVoice was already trained with ASR prediction. While CosyVoice used larger codebook size, the paper only used smaller codebook size for information bottleneck. While the approach is effective for speech disentanglement, it does not introduce any novelty for research paper.

Furthermore, the proposed destylizer is trained only on English-only dataset, resulting in higher performance on English test set. It would be beneficial to include multi-lingual evaluation to assess the generalization performance.

[DiT Decoder]
There already exist numerous DiT and CFM-based VC models. However, the paper do not cite or discuss any of them. TBH, I do not want to waste my time to review this paper. Please examine related works carefully. You can find similar works published at ICASSP and Interspeech.

[Streaming Design]
The use of causal Layer and ring buffer has already been proposed in previous work [1].

[1] Yang, Yang, et al. "Streamvc: Real-time low-latency voice conversion." ICASSP 2024-2024 IEEE International Conference on Acoustics, Speech and Signal Processing (ICASSP). IEEE, 2024.

[Overclaimed statement]

I strongly disagree with the statement in line 69. The paper only employs a style encoder that do not control the timbre, accent, and emotion separately. If such claims are to be made, please add the experiment of the disentangled control for these attributes (timbre, accent, and emotion). Please avoid overstating the contributions of the paper.

All models trained on various datasets can jointly mimic these attributes from the reference speech. However, achieving disentangled control over them remains challenging.

**Questions:**

Q1. Please add the sampling steps and sampling methods for CFM decoder.

Q2. Do you really consider your model to be a real-time system with a latency of 1 second?

Q3. Which frame rate did you use for the FSQ tokens? In line 179, it states, "Unlike CosyVoice2, where a large codebook size (6561) is used, we use a much smaller codebook to enforce a narrower information bottleneck." This implies that the paper used 25 Hz for FSQ. However, if the model actually used 50 Hz for FSQ, this statement appears to be an overclaim.

**Details Of Ethics Concerns:**

.

---

> ### Author Response · Authors · 2025-11-23
> **Response to Reviewer EHyw (Part 1 / 2)**
>
> Thank you for your review. Here we address your concerns as follows:
>
> >my primary concern lies in the lack of novelty
>
> As we mentioned in the common response, we respectfully argue that our work addresses a fundamental **representation learning** challenge. While we leverage established modules (ASR supervision, FSQ), our core novelty lies in discovering a specific **disentangled representation** that standard approaches fail to capture.
>
> While our destylizer builds on top of the CosyVoice 2, we diverge from standard practices in two critical ways:
>
> **1. Information Bottleneck:** Unlike CosyVoice 2, which uses large codebooks that inadvertently encode style, we identify that a highly compact codebook (size 45) is essential for stripping away style information.
>
> **2. Continuous Feature Space:** We propose the use of **pre-quantization continuous features** rather than discrete tokens. Our ablation studies (Section 5.4) demonstrate that this design choice is not merely a hyperparameter tune, but a structural requirement for achieving clean content-style disentanglement without sacrificing intelligibility.
>
> Therefore, our novelty is the discovery of this specific **disentangled content representation**. Crucially, this robust disentanglement eliminates the need for complex autoregressive error correction, directly enabling the **first streamable zero-shot VSC framework**.
>
> > There already exist numerous DiT and CFM-based VC models.
>
> First, we would like to restate the definition of **Voice Style Conversion (VSC)**: generating speech that preserves the source content while transferring the target’s voice style, specifically **timbre, accent, and emotion**, which we defined in the introduction of our paper on lines 32-33.
> While literature exists for individual voice style attributes, such as **Timbre** (Seed-VC [1]), **Accent** (ConvSpeak [2], TokAN [3]), or **Emotion** (EmoVox [4], PDSN [5]), these approaches face distinct limitations, including reliance on paired data (ConvSpeak), not many-to-many (TokAN), or dependency on external style labels and classifier (PDSN, EmoVox). Crucially, while prior work exists for streaming voice conversion (timbre-only), no existing framework tackles **streaming VSC** (joint timbre, accent, and emotion transfer).
>
> Currently, Vevo (ICLR 2025) [6] is **the only prior work** capable of offline VSC (timbre, accent and emotion). Furthermore, Vevo faces two critical limitations:
>
> **1. Suboptimal Disentanglement:** As shown in our analysis (Section 5.3), Vevo’s content tokens suffer from style leakage and degraded content intelligibility.
>
> **2. Streaming Incompatibility:** Vevo relies on an LLM (autoregressive transformer) for duration modeling. The inherent input-output length mismatch in LLM-based generation makes real-time speech-to-speech streaming fundamentally difficult.
>
> While prior work such as StreamVC [7] have explored architectural advancements to enable streaming for VC, we argue that our main contribution is in the development of a streamable and disentangled content representation using our destylizer, which allows us to apply these techniques for a streaming VSC system.
>
> [1] Liu, S. (2024). Zero-shot voice conversion with diffusion transformers. arXiv preprint arXiv:2411.09943.
>
> [2] Xue, H., Peng, X., & Lu, Y. (2024). Convert and speak: Zero-shot accent conversion with minimum supervision. In ACM Multimedia 2024.
>
> [3] Bai, Q., Inoue, S., Wang, S., Jiang, Z., Wang, Y., Li, H. (2025) Accent Normalization Using Self-Supervised Discrete Tokens with Non-Parallel Data. Proc. Interspeech 2025
>
> [4] Zhou, K., Sisman, B., Rana, R., Schuller, B. W., & Li, H. (2022). Emotion intensity and its control for emotional voice conversion. IEEE Transactions on Affective Computing
>
> [5] Jia, D., Tian, Q., Peng, K., Li, J., Chen, Y., Ma, M., Wang, Y., Wang, Y. (2023) Zero-Shot Accent Conversion using Pseudo Siamese Disentanglement Network. Proc. Interspeech 2023
>
> [6] Zhang, X., Zhang, X., Peng, K., Tang, Z., Manohar, V., Liu, Y., ... & Ma, M. (2025). Vevo: Controllable zero-shot voice imitation with self-supervised disentanglement. ICLR 2025
>
> [7] Yang, Y., Kartynnik, Y., Li, Y., Tang, J., Li, X., Sung, G., & Grundmann, M. (2024, April). Streamvc: Real-time low-latency voice conversion. In ICASSP 2024-2024 IEEE International Conference on Acoustics, Speech and Signal Processing (ICASSP)

---

> ### Author Response · Authors · 2025-11-23
> **Response to Reviewer EHyw (Part 2 / 2)**
>
> >It would be beneficial to include multi-lingual evaluation to assess the generalization performance.
>
> Our study primarily focuses on English to ensure a rigorous objective evaluation of accent and emotion cloning. Stylistic attributes are often language-dependent (e.g., tonal patterns in Mandarin vs. stress patterns in English), and standardized automatic accent similarity metrics for non-English languages are currently lacking.
>
> However, to address the reviewer's interest in multi-lingual evaluation, we conducted an additional **zero-shot cross-lingual VSC** experiment using our current model. We used English utterances as the source and non-English utterances (Spanish, French, Italian, Hindi, Arabic, and Russian) as the target. We found that even without explicit multilingual training, StyleStream effectively transfers the voice style of the foreign speakers onto the English content. We have updated our demo page (https://stylestream10033.github.io/stylestream/) with these samples to demonstrate this capability.
>
> >I strongly disagree with the statement in line 69. Please avoid overstating the contributions of the paper.
>
> To clarify, on line 69 we wrote that “there is no existing work that addresses real-time voice style conversion (timbre, accent and emotion), leaving an important gap.” We do not ever claim that we have “disentangled” control over these features in our current work. We only claim that we are able to convert these three attributes of the speech in a real-time manner. Furthermore, we would like to emphasize that our system is the **first streamable zero-shot VSC framework**.
>
> >Q1: Please add the sampling steps and sampling methods for CFM decoder.
>
> We thank the reviewer for pointing this out. For the CFM decoder, we employ the standard Euler ODE solver with the Number of Function Evaluations (NFE) set to 16 across all evaluations. We have updated Section 4.2 in the revised paper to explicitly state these hyperparameters.
>
> >Q2: Do you really consider your model to be a real-time system with a latency of 1 second?
>
> Yes, we consider the system to be a real-time system with a latency of 1 second. Our system consumes the input in fixed chunk intervals and processes them faster than the length of the chunks, so it is by definition a streaming, real-time system. Please see our video demo that we have added at https://stylestream10033.github.io/stylestream/ where we show that the system runs in a realtime fashion on-device.
>
> >Q3: Which frame rate did you use for the FSQ tokens?
>
> We clarify that the frame rate for our FSQ tokens is 50 Hz. However, we respectfully point out that our claim of a "narrower information bottleneck" holds true when analyzing the **Total Information Bitrate** and is also empirically verified by our ablation studies.
>
> **1. Theoretical Bottleneck (Bitrate Analysis):** The information capacity is determined by both frame rate ($R$) and codebook size ($N$). The bitrate is calculated as $R \times \log_2(N)$.
>
> - CosyVoice 2: $25 \text{ Hz} \times \log_2(6561) \approx \mathbf{317.0 \text{ bits/s}}$.
> - StyleStream (Ours): $50 \text{ Hz} \times \log_2(45) \approx \mathbf{274.5 \text{ bits/s}}$.
>
> Thus, despite the higher frame rate, our system enforces a ~13% tighter information constraint than CosyVoice 2.
>
> **2. Empirical Verification (Style Leakage):** Crucially, this theoretical constraint is supported by our experiments.
>
> - CosyVoice 2 Leakage: As shown in our analysis **(Section 5.3)**, features extracted using CosyVoice 2’s large codebook exhibit high correlation with style attributes, indicating severe style leakage.
>
> - Codebook Ablation: Our ablation study **(Section 5.4)** explicitly compares codebook sizes. We demonstrate that increasing the size beyond 45 reintroduces style information, degrading disentanglement.
>
> Therefore, the combination of the compact codebook (size 45) and ASR supervision creates a functionally narrower bottleneck for style information, regardless of the 50 Hz frame rate.

---

> > ### Comment · Reviewer_EHyw · 2025-11-25
> >
> > Thanks for your response.
> >
> > After reading the responses from authors, I am confident that the authors are not experts in speech synthesis.
> >
> > All models using global speaker encoder can mimic the timbre, accent, and emotion from a reference audio. It is very natural results and should not be claimed as a contribution of the paper. I strongly recommend reviewing classic papers on style modeling [1], [2]. Historically, many studies have focused on disentangled control of each speech components, rather than the entire style.
> >
> > [1] Skerry-Ryan, R. J., et al. "Towards end-to-end prosody transfer for expressive speech synthesis with tacotron." international conference on machine learning. PMLR, 2018.
> >
> > [2] Wang, Yuxuan, et al. "Style tokens: Unsupervised style modeling, control and transfer in end-to-end speech synthesis." International conference on machine learning. PMLR, 2018.
> >
> > Also,
> >
> > Regarding the zero-shot cross-lingual VSC,
> >
> > I do not request the authors to perform cross-lingual VSC using an English source speech. Since the current tokens are trained using only English dataset, the comparison against CosyVoice 2.0 or other models which were trained with Enlgish and Chinese is unfair. Please try to convert the style using other language source speech, not English. Also, please summarize the evaluation table with other models.
> >
> > Furthermore, I respectfully disagree the statement
> > - "Unlike CosyVoice2, where a large codebook size (6561) is used, we use a much smaller codebook to enforce a narrower information bottleneck."
> >
> > as the tokens are quantized at 50 Hz, whereas CosyVoice uses 25 Hz. This may mislead readers and should be corrected by stating the real bitrate.
> >
> > Finally, the audio quality of real-time demo is too poor and far from the practical application. I recommend examining commercially deployed systems such as Elevenlabs (voice cloning) or Supertone (shift) to understand the expected quality for real-time applications.

---

> > > ### Comment · Reviewer_EHyw · 2025-11-25
> > >
> > > While StyleStream cannot independently control each style component, Vevo is indeed a more advanced model, as it can transfer accent and timbre separately.
> > >
> > > Please do not oversell your paper.

---

> > > ### Author Response · Authors · 2025-11-27
> > > **2nd Round Response to Reviewer EHyw (Part 1 / 2)**
> > >
> > > >All models using global speaker encoder can mimic the timbre, accent, and emotion from a reference audio.
> > >
> > > We thank the reviewer for referencing these foundational works in TTS. While global speaker encoders are indeed sufficient for **TTS** (where the input text is stylistically neutral and inherently disentangled from speaking style), they are fundamentally insufficient for **VSC** (where the input audio carries strong intrinsic style), which is the stated goal of this work.
> > >
> > > This distinction creates an **"Entanglement Gap"**: applying a global encoder directly to standard acoustic features results in **style interference**, as the source’s intrinsic accent and emotion clash with the target attributes. This is not just theoretical; it is quantified in our **style leakage analysis (Section 5.3)**, which demonstrates that entangled features retain substantial source style information, preventing effective conversion.
> > >
> > > Therefore, our core contribution is not the global encoder itself, but the **Destylizer**. By designing a mechanism that successfully "neutralizes" the input audio into a disentangled, text-like, and streamable representation, we enable the global stylization pipeline to function reliably for speech-to-speech tasks, achieving the clean transfer that standard encoders have not demonstrated so far.
> > >
> > > >Please try to convert the style using other language source speech, not English.
> > >
> > > We thank the reviewer for the follow-up. We respectfully clarify two points regarding scope and comparison fairness.
> > >
> > > **1. Feasibility of Non-English Source Speech:** Evaluating non-English source speech is **architecturally infeasible and outside the scope** of our paper. Our semantic backbone relies on English-specific extractors. Feeding non-English audio into these modules would result in semantic collapse (invalid content features), making conversion impossible regardless of the model's capabilities.
> > >
> > > We firmly believe our rigorous experiments in the English domain are sufficient to validate our core contributions: the **Destylizer** and the **first streaming VSC framework**. Extending this to multilingual data is not merely an engineering task but a distinct, open research topic. Multilingual Voice Style Conversion requires discovering a **language-independent, style-agnostic content latent space**, which necessitates a fundamental redesign of the information bottleneck and stylization mechanisms to handle cross-lingual phonological shifts. Solving this complex representation learning challenge is a substantial undertaking that is orthogonal to the contribution of this work.
> > >
> > >
> > > **2. Fairness of CosyVoice 2 Comparison:** Regarding training data, CosyVoice 2 is the only baseline model that is trained with multilingual data. However, we emphasize that CosyVoice 2's poor performance in VSC is **architectural, not linguistic**. Its failure stems from the **information bottleneck size**, not the training languages. CosyVoice 2 utilizes a massive codebook (6,561 tokens). As demonstrated in our **Style Leakage Analysis (Table 5)** and **Bottleneck Ablation (Table 7)**, larger codebooks inevitably encode source style alongside content, leading to poorer style similarity (high leakage). Therefore, CosyVoice 2 fails the VSC task due to this fundamental design choice.
> > >
> > > >This may mislead readers and should be corrected by stating the real bitrate.
> > >
> > > We respectfully refer the reviewer to our **Round 1 Response**, where we provided a detailed **Bitrate Analysis**. We demonstrated mathematically that despite the 50Hz frame rate, StyleStream enforces a **~13% tighter information constraint** (in bits/s) than CosyVoice 2.
> > >
> > > Furthermore, our claim is not just theoretical but **empirically verified**. As shown in our style leakage analysis **(Section 5.3)** and codebook ablation studies **(Table 7)**, the compact codebook (size 45) prevents the encoding of paralinguistic details that the large CosyVoice codebook (size 6561) retains. Therefore, the statement accurately reflects both the information theoretic bound and the observed disentanglement behavior. To avoid ambiguity, we will explicitly mention the frame rates and resulting bitrates in the final revision.

---

> > > > ### Author Response · Authors · 2025-11-27
> > > > **2nd Round Response to Reviewer EHyw (Part 2 / 2)**
> > > >
> > > > >Finally, the audio quality of real-time demo is too poor and far from the practical application
> > > >
> > > > We respectfully note that ElevenLabs and Supertone are **proprietary commercial products**, not reproducible scientific baselines. Their architectures, training data scale, and computational budgets are undisclosed, making a scientific comparison impossible.
> > > >
> > > > Per standard academic practice, we compare against **state-of-the-art peer-reviewed methods** (Vevo, CosyVoice 2). As shown in **Table 1**, StyleStream outperforms these baselines across all metrics. Furthermore, StyleStream is the **first academic work** to demonstrate **streaming** zero-shot voice style conversion (timbre, accent and emotion). We believe our demo validates the feasibility of this novel architecture on consumer hardware (RTX 4060), representing a significant leap over existing peer-reviewed baselines, even if it has not yet reached the polish of mature commercial engines backed by large engineering teams.
> > > >
> > > > >Please do not oversell your paper
> > > >
> > > > We respectfully clarify that we do not claim independent control over individual style attributes. We fully acknowledge Vevo's capability for attribute-specific manipulation. However, our comparison focuses on **Holistic Voice Style Conversion (VSC)**, the task of jointly transferring the target's complete vocal identity (Timbre + Accent + Emotion) to the source content.
> > > >
> > > > In this specific context (Vevo-Voice setting), our claims are strictly based on empirical evidence, not "overselling":
> > > >
> > > > - **Significantly improved Fidelity:** As shown in **Table 1**, StyleStream significantly outperforms Vevo in holistic similarity metrics (**S-SIM**: 0.855 vs 0.818; **A-SIM**: 0.635 vs 0.596; **E-SIM**: 0.803 vs 0.712). This indicates that while Vevo offers granular control, StyleStream achieves higher fidelity in capturing the global target style.
> > > >
> > > > - **Streaming Capability:** We effectively trade the granular controllability of Vevo for the capability of **real-time streaming**. While Vevo only works offline, StyleStream solves the harder challenge of achieving high-fidelity holistic conversion within a streaming constraint.
> > > >
> > > > Therefore, we believe describing StyleStream as state-of-the-art for **Real-Time VSC** is an accurate reflection of the experimental results.

---

### Official Review · Reviewer_FPmA · 2025-11-02

**Soundness:** 3
**Presentation:** 3
**Contribution:** 3
**Rating:** 6
**Confidence:** 4

**Summary:**

The paper introduces StyleStream, a framework for real-time zero-shot voice style conversion capable of transforming an input utterance’s timbre, accent, and emotion to match a target speaker’s style, while preserving the original linguistic content. The proposed system combines a destylizer trained with ASR supervision and a compact finite scalar quantization (FSQ) bottleneck to achieve cleaner content–style disentanglement, and a diffusion transformer stylizer for style reconstruction. The system achieves ~1s end-to-end latency, representing a significant advance toward practical real-time style conversion. Experimental results show that StyleStream outperforms previous methods (CosyVoice 2.0, Vevo, SeedVC) across multiple style similarity and MOS metrics.

**Strengths:**

1. Real-time voice style conversion is an important and challenging problem that requires transferring not only timbre but also higher-level stylistic cues (accent and emotion). Tackling this in a streaming setting is timely and practically valuable.
2. Experimental results are strong, and the demo samples sound convincing, with clear style transfer and good intelligibility.
3. The method design is well thought out: combining ASR loss with a small quantization codebook effectively improves the destylizer’s disentanglement ability. The use of continuous pre-quantization features is a useful empirical trick that improves performance and is well justified by ablation studies.

**Weaknesses:**

1. The main contribution lies in the task and empirical achievement—a functioning real-time voice style conversion system—rather than in methodological novelty. The ASR-supervised tokenizer and DiT-based spectrogram generator follow ideas already seen in recent works (e.g., CosyVoice2, E2-TTS, F5-TTS). Hence, the paper’s conceptual originality for the machine learning community may be somewhat limited, though it is impactful for speech research.

2. A stronger baseline would help: a simple cascaded ASR + zero-shot TTS pipeline could demonstrate more clearly where StyleStream’s advantages lie (e.g., latency, style fidelity).

3. The experimental scope could be expanded. Multilingual and higher-sampling-rate (e.g., 24 kHz) experiments would make the evaluation more solid and test the model’s generalization.

**Questions:**

1. In Table 1, for CosyVoice 2, Vevo, and Vevo 1.5, did the authors use only the flow-matching (FM) component for zero-shot conversion, or the full AR + FM pipeline? My concern is that using only the FM module may limit those baselines to timbre conversion, not full style transfer.
2. Regarding the streaming design, how exactly is DiT modified to support causal or chunked inference? Is the approach similar to that of CosyVoice 2?
3. The training data split is somewhat unclear: the destylizer and stylizer use different datasets. Could the destylizer benefit from being trained on the same large-scale dataset as the stylizer (e.g., Emilia)? What are the constraints preventing this?

---

> ### Author Response · Authors · 2025-11-23
> **Response to Reviewer FPmA (Part 1 / 3)**
>
> Thank you for your insightful review. We are encouraged that you think our work is "timely and practically valuable". Here we will address your concerns as follows:
>
> >Hence, the paper’s conceptual originality for the machine learning community may be somewhat limited, though it is impactful for speech research.
>
> We appreciate the recognition of our system's empirical success. However, we respectfully argue that our contribution extends beyond system integration and addresses a fundamental **representation learning** challenge. While we leverage established modules (ASR supervision, FSQ), our core novelty lies in discovering a specific **disentangled representation** that standard approaches fail to capture.
>
> Specifically, we diverge from standard practices in two critical ways:
>
> **1. Information Bottleneck:** Unlike CosyVoice 2, which uses large codebooks that inadvertently encode style, we identify that a highly compact codebook (size 45) is essential for stripping away style information.
>
> **2. Continuous Feature Space:** We propose the use of **pre-quantization continuous features** rather than discrete tokens. Our ablation studies (Section 5.4) demonstrate that this design choice is not merely a hyperparameter tune, but a structural requirement for achieving clean content-style disentanglement without sacrificing intelligibility.
>
> Therefore, our novelty is the discovery of this specific **disentangled content representation**. Crucially, this robust disentanglement eliminates the need for complex autoregressive error correction, directly enabling the **first streamable zero-shot VSC framework**.
>
> >A stronger baseline would help: a simple cascaded ASR + zero-shot TTS pipeline could demonstrate more clearly where StyleStream’s advantages lie (e.g., latency, style fidelity).
>
>  We appreciate the suggestion to benchmark against a stronger baseline. We have constructed an offline cascaded ASR+TTS pipeline using **Whisper-large-v3** and **CosyVoice 2.0**. The comparative results are presented below:
>
> | Model                   | WER (%) ↓ | S-SIM ↑ | A-SIM ↑ | E-SIM ↑ |
> | :---                    | :---:     | :---:   | :---:   | :---:   |
> | StyleStream (streaming) | 15.3      | 0.852   | 0.635   | 0.803   |
> | StyleStream (offline)   | 9.2       | 0.856   | 0.640   | 0.827    |
> | ASR+TTS (offline)        | 7.4       | 0.840   | 0.624   | 0.740   |
>
> In the offline setting, the cascaded baseline is comparable to StyleStream. While the ASR+TTS baseline achieves a slightly lower WER (benefiting from explicit text input to TTS, which enforces content stability provided the ASR transcription is accurate), StyleStream demonstrates superior performance in style fidelity. Specifically, StyleStream achieves higher similarity scores across timbre, accent, and emotion.
>
> However, we emphasize that while ASR+TTS is viable offline, it faces fundamental structural limitations in **streaming settings**:
>
> **1. Input-Output Synchronization & Duration Mismatch:** The cascaded baseline relies on text as an intermediate representation, which inherently discards **duration** information. Consequently, the autoregressive TTS generates speech with durations uncorrelated to the input source, leading to significant **synchronization drift** (overlap or heavy delays) that breaks the streaming pipeline. In contrast, StyleStream is designed with a fully non-autoregressive architecture, ensuring input-output alignment is maintained naturally during streaming.
>
> **2. Latency & Error Propagation:** A cascaded streaming system is bottlenecked by the ASR module's latency and stability. The TTS module risks starvation while waiting for the ASR to commit to uncertain tokens, causing audio glitches or compounded latency. Furthermore, any ASR transcription errors permanently corrupt the TTS output. StyleStream avoids this by utilizing continuous content features from the streaming Destylizer. This allows the model to adapt to acoustic input continuously without the latency penalties or hard-threshold errors associated with discrete text tokenization.

---

> > ### Author Response · Authors · 2025-11-23
> > **Response to Reviewer FPmA (Part 2 / 3)**
> >
> > >Multilingual and higher-sampling-rate (e.g., 24 kHz) experiments would make the evaluation more solid and test the model’s generalization.
> >
> > **1. Multilingual Generalization:** Our study primarily focuses on English to ensure a rigorous objective evaluation of accent and emotion cloning. Stylistic attributes are often language-dependent (e.g., tonal patterns in Mandarin vs. stress patterns in English), and standardized automatic accent similarity metrics for non-English languages are currently lacking.
> >
> > However, to address the reviewer's interest in generalization, we conducted an additional **zero-shot cross-lingual VSC** experiment using our current model. We used English utterances as the source and non-English utterances (Spanish, French, Italian, Hindi, Arabic, and Russian) as the target. We found that even without explicit multilingual training, StyleStream successfully transfers the voice style of the foreign speakers onto the English content. We have updated our demo page (https://stylestream10033.github.io/stylestream/) with these samples to demonstrate this capability.
> >
> > **2. 24 kHz Sampling Rate:** Regarding the suggestion for higher sampling rates, we note that the output resolution is primarily determined by the vocoder backend (e.g., training Vocos at 24 kHz) rather than the core StyleStream architecture. As scaling to 24 kHz is a standard engineering integration orthogonal to our primary contributions in streaming mechanics and disentanglement, we believe the current evaluation fully validates the proposed method.
> >
> > >Q1: In Table 1, for CosyVoice 2, Vevo, and Vevo 1.5, did the authors use only the flow-matching (FM) component for zero-shot conversion, or the full AR + FM pipeline?
> >
> > We utilized the official inference implementations for all baselines. However, the inference pipelines differ based on the architecture of each model:
> >
> > **1. Vevo and Vevo 1.5:** We utilized the **full pipeline (LLM + DiT)**. For these models, the official voice style conversion pipeline involves extracting content tokens from the source audio, passing them through the LLM to generate content-style tokens, and finally using the DiT for waveform generation. Because the LLM is involved in the intermediate stage to model the target style, these baselines perform **full voice style conversion**, not just timbre conversion.
> >
> > **2. CosyVoice 2:** We utilized the **Flow Matching/DiT component only**. We clarify that the official CosyVoice 2 architecture can just support **timbre-only voice conversion**. It does not support token-to-token translation via its LLM, since its LLM is designed strictly for text-to-token. Consequently, the official pipeline bypasses the LLM: it extracts speech tokens from the source and feeds them directly into the DiT, conditioned on the target speaker embedding and context mel-spectrogram. We acknowledge the reviewer’s insight that this setup primarily performs timbre transfer, because the source speech tokens in CosyVoice 2 are not fully disentangled from the source style, as shown in Section 5.3.
> >
> > > Q2: Regarding the streaming design, how exactly is DiT modified to support causal or chunked inference? Is the approach similar to that of CosyVoice 2?
> >
> > Yes, the approach shares similarities with CosyVoice 2 regarding the masking strategy. As detailed in **Section 3.4**, we train the DiT using a **chunked causal attention mask** to enforce causality. For the inference stage **(Section 5.2)**, we maintain a **5-second content ring buffer**. This buffer feeds the DiT the necessary historical context for the current chunk, ensuring smooth transitions and stability during streaming generation.

---

> > > ### Author Response · Authors · 2025-11-23
> > > **Response to Reviewer FPmA (Part 3 / 3)**
> > >
> > > > Q3: Could the destylizer benefit from being trained on the same large-scale dataset as the stylizer (e.g., Emilia)?
> > >
> > > We thank the reviewer for the suggestion regarding data scaling. To investigate this, we conducted experiments where we trained the Destylizer on the Emilia-EN dataset and retrained the Stylizer on top of it. The comparative results are shown below:
> > >
> > > | Training data                                 | WER (%) ↓ | S-SIM ↑ | A-SIM ↑ | E-SIM ↑ |
> > > | :---                                          | :---:     | :---:   | :---:   | :---:     |
> > > | LibriTTS + MSP-Podcast + GLOBE (1300 hours)   | 10.7      | 0.837   | 0.626   | 0.733   |
> > > | Emilia (50k hours)                            | 10.3      | 0.816   | 0.594   | 0.687  |
> > >
> > > As shown, scaling the Destylizer to 50k hours yields **no significant overall improvement**. While there is a marginal gain in content preservation (WER drops from 10.7% to 10.3%), the style disentanglement metrics (S-SIM, A-SIM) actually show a slight degradation. We attribute this saturation to two factors:
> > >
> > > **1. Task Nature (Discriminative vs. Generative)**: The Destylizer performs a **discriminative extraction task** (compressing speech into content features via ASR supervision), which differs fundamentally from the Stylizer's generative task. While generative models (DiT) require massive data to capture long-tail acoustic distributions, the Destylizer's task saturates much faster. Once the model learns robust content-style disentanglement, adding more data yields diminishing returns.
> > >
> > > **2. Strategic Data Sufficiency**: Our 1.3k-hour dataset was constructed to maximize **style variance**, not just total duration. By combining **GLOBE** (high accent diversity), **MSP-Podcast** (high emotion diversity), and **LibriTTS** (clean timbre), this dataset already covers the necessary manifold of style variations required for the bottleneck to learn disentanglement. Adding the massive "in-the-wild" Emilia data does not significantly expand this style manifold relative to the Destylizer's needs, and as observed, may even introduce noise that slightly hampers style isolation.
> > >
> > > Therefore, we conclude that the 1.3k-hour dataset is the optimal efficiency point for the Destylizer.

---

### Author Response · Authors · 2025-11-23
**Common Response: Novelty, Contributions, and Comparison with Prior Work (Part 1 / 2)**

We sincerely thank the reviewers for their insightful feedback and for recognizing StyleStream as a "highly original and technically impressive contribution" (Reviewer HLfV) that addresses a "genuinely new" and "practically valuable" problem (Reviewer FPmA). We are encouraged that the reviewers validated our specific architectural choices as a "clever hybridization" (HLfV) and "well thought out" (FPmA), supported by "well-designed experiments" (Reviewer MYRv) that confirm our system successfully balances high-quality disentanglement with real-time efficiency.

To ensure clarity in the following discussions, we restate the definition of **Voice Style Conversion (VSC)**: generating speech that preserves the source content while transferring the target’s voice style, specifically **timbre, accent, and emotion**. Within this scope, our core contribution is twofold:

1. **First Streamable VSC**: We propose the first framework capable of zero-shot VSC in a real-time, streaming setting. Unlike prior works that are restricted to **single-attribute conversion** (e.g., timbre-only) or **offline inference**, StyleStream achieves holistic, real-time VSC.

2. **Representation Insight**: We provide rigorous empirical studies justifying the design of the **Destylizer (Section 5.3 & 5.4)**, demonstrating that our specific configuration (ASR + compact codebook + pre-quantization continuous features) delivers significantly cleaner content-style disentanglement than prior state-of-the-art approaches.

We strongly encourage reviewers to view our **anonymous video demo** (https://stylestream10033.github.io/stylestream/), which showcases the system performing **real-time voice style conversion** on a local laptop with an RTX 4060 GPU.

We elaborate on the novelty of these contributions and address specific comparisons with prior work below:

**1. Limitations of Prior Work**

While literature exists for individual voice style attributes, such as **Timbre** (Seed-VC [1]), **Accent** (ConvSpeak [2], TokAN [3]), or **Emotion** (EmoVox [4], PDSN [5]), these approaches face distinct limitations, including reliance on paired data (ConvSpeak), not many-to-many (TokAN), or dependency on external style labels and classifier (PDSN, EmoVox). Crucially, while prior work exists for streaming voice conversion (timbre-only), no existing framework tackles **streaming VSC** (joint timbre, accent, and emotion transfer).

Currently, **Vevo (ICLR 2025) [6]** is the only prior work capable of offline VSC (timbre, accent and emotion). Furthermore, Vevo faces two critical limitations:
- **Suboptimal Disentanglement**: As shown in our analysis **(Section 5.3)**, Vevo’s content tokens suffer from style leakage and degraded content intelligibility.
- **Streaming Incompatibility**: Vevo relies on an LLM (autoregressive transformer) for duration modeling. The inherent input-output length mismatch in LLM-based generation makes real-time speech-to-speech streaming fundamentally difficult.

**2. Destylizer vs. CosyVoice 2**

Regarding the Destylizer’s architectural similarity to CosyVoice 2 [7], we emphasize that our contribution addresses a fundamental **representation learning challenge**. While CosyVoice 2 employs standard large codebooks that lead to significant style leakage (as evidenced in **Table 5**), we identify a critical representational insight: a **compact codebook (size 45)** combined with **continuous pre-quantization features** creates the necessary information bottleneck to strip away style while preserving content. This design is not arbitrary; it is rigorously validated by our ablation studies **(Tables 6 & 7)**. We believe the in-depth analysis of style leakage **(Section 5.3)** and architectural design **(Section 5.4)** offers valuable, generalizable insights to the speech community regarding robust content-style disentanglement.

---

> ### Author Response · Authors · 2025-11-23
> **Common Response: Novelty, Contributions, and Comparison with Prior Work (Part 2 / 2)**
>
> **3. From Disentanglement to Streaming**
>
> This improved disentanglement is the key to unlock streaming. Because our features are robustly disentangled, StyleStream eliminates the need for LLMs to handle duration modeling or content correction (unlike Vevo). Instead, we rely solely on a fully non-autoregressive DiT. This avoids the I/O length mismatch typical of LLM-based approaches, making streaming a **natural outcome** of our architecture rather than an engineered add-on.
>
> We have uploaded a revised manuscript with changes highlighted in red, incorporating feedback from the reviewers. Additionally, during a rigorous re-verification of our evaluation, we corrected an implementation inaccuracy in the speaker similarity (S-SIM) evaluation. The updated results reflect this correction, revealing that StyleStream achieves even higher style fidelity than originally reported, further solidifying its state-of-the-art performance against baselines.
>
>
>
> [1] Liu, S. (2024). Zero-shot voice conversion with diffusion transformers. arXiv preprint arXiv:2411.09943.
>
> [2] Xue, H., Peng, X., & Lu, Y. (2024). Convert and speak: Zero-shot accent conversion with minimum supervision. In ACM Multimedia 2024.
>
> [3] Bai, Q., Inoue, S., Wang, S., Jiang, Z., Wang, Y., Li, H. (2025) Accent Normalization Using Self-Supervised Discrete Tokens with Non-Parallel Data. Proc. Interspeech 2025
>
> [4] Zhou, K., Sisman, B., Rana, R., Schuller, B. W., & Li, H. (2022). Emotion intensity and its control for emotional voice conversion. IEEE Transactions on Affective Computing
>
> [5] Jia, D., Tian, Q., Peng, K., Li, J., Chen, Y., Ma, M., Wang, Y., Wang, Y. (2023) Zero-Shot Accent Conversion using Pseudo Siamese Disentanglement Network. Proc. Interspeech 2023
>
> [6] Zhang, X., Zhang, X., Peng, K., Tang, Z., Manohar, V., Liu, Y., ... & Ma, M. (2025). Vevo: Controllable zero-shot voice imitation with self-supervised disentanglement. ICLR 2025
>
> [7] Du, Z., Wang, Y., Chen, Q., Shi, X., Lv, X., Zhao, T., ... & Zhou, J. (2024). Cosyvoice 2: Scalable streaming speech synthesis with large language models. arXiv preprint arXiv:2412.10117.

---

### Meta-Review · Area_Chair_vo9Q · 2026-01-09

**Summary:**

The paper proposes StyleStream, a framework for real-time zero-shot voice style conversion that attempts to transfer timbre, accent, and emotion simultaneously. The authors employ a destylizer with ASR supervision and a finite scalar quantization bottleneck to disentangle content, followed by a diffusion transformer for stylization. While some reviewers recognized the practical value and the specific architectural choices enabling streaming, there was significant disagreement regarding the novelty and scientific contribution of the work. Two reviewers strongly criticized the paper for lacking conceptual novelty and relying heavily on the engineering integration of existing modules. Despite the authors' rebuttal addressing the distinction between their work and prior timbre-only conversion, the consensus reflects that the contributions are somewhat incremental relative to the broader machine learning community. Given the highly polarized scores and the substantial concerns regarding the depth of the methodological contribution versus engineering effort, the paper is not recommended for acceptance at this time.

**Reviewer Concerns:**

The primary outstanding concern shared by multiple reviewers is the limited methodological novelty, with critics arguing that the system represents an engineering integration of well-known components (ASR supervision, FSQ, DiT) rather than a fundamental advancement in representation learning. While the authors successfully clarified the distinction between their holistic style conversion task and standard timbre-only voice conversion during the rebuttal, concerns persist regarding the definition and competitiveness of the "real-time" claim. Specifically, reviewers noted that the 1-second latency is high compared to other real-time systems and questioned the practical deployment on consumer hardware, although the authors did provide an RTX 4060 demo in response. Additionally, the restriction to English-only data was flagged as a limitation for a general voice style transfer system, and while the authors provided zero-shot cross-lingual examples, the lack of rigorous multilingual training remains a limitation. The rebuttal addressed technical clarifications regarding the FSQ bottleneck and baseline comparisons, but the core skepticism regarding the sufficient novelty for ICLR remained among the more critical reviewers.

**Reviewer Scores:**

Reviewers FPmA and HLfV (initial scores of 6) acknowledged the rebuttal's clarifications on baselines and streaming mechanisms and would likely maintain their borderline positive assessment or potentially shift slightly depending on the weight given to the novelty concerns raised by others. Reviewer MYRV (score 4) might have slightly improved their perception based on the additional real-time benchmarks and consumer hardware demo, but likely remains borderline due to the persistent latency comparisons with faster systems. Reviewers EHyw and UdFS (scores of 0) expressed strong fundamental objections to the paper's novelty and contribution; while their scoring was exceptionally harsh and flagged by the authors as unprofessional outliers, it is unlikely they would have moved their scores significantly above the rejection threshold, as they viewed the work fundamentally as engineering rather than research.

---

### Decision · Program_Chairs · 2026-01-26

Reject